# Noncanonical roles of chemokine regions in CCR9 activation revealed by structural modeling and mutational mapping

Inês De Magalhaes Pinheiro[1,4], John R. D. Dawson[2,4], Nicolas Calo [1,3], Marianne Paolini-Bertrand[1], Kalyana Bharati Akondi[1,4], Gavin Tan[2], Tracy M. Handel [2], Irina Kufareva [2] ✉ & Oliver Hartley [1,3] ✉

The G protein-coupled chemokine receptor CCR9 plays a major role in inflammatory bowel disease and is implicated in cancer. Despite its therapeutic relevance, the mechanism by which CCR9 is activated by its endogenous chemokine CCL25 remains poorly understood. Here, we combine structural modeling with multimodal pharmacological analysis of CCR9 mutants to map the CCR9–CCL25 interface and delineate key determinants of binding, G protein versus arrestin signaling, and constitutive activity. We show that unlike other chemokines which drive receptor activation through their N-termini, CCL25 activates CCR9 via a distinct region, its 30s loop. Supporting this non-canonical mechanism, CCR9 signaling tolerates alanine mutations in the CCL25 N-terminus but is strongly affected by 30s loop modifications. Engineered N-terminally modified CCL25 analogs remain full agonists, consistent with signaling determinants lying outside the N-terminus. This non-canonical activation signature provides insights for CCR9 drug discovery and may inform structure-based design for other chemokine receptors.

Chemokine receptors are members of the G protein-coupled (GPCR) chemokine receptor superfamily with a principal role in controlling the activation and trafficking of leukocytes[1]. They have been identified as key players in inflammation, infectious diseases and cancer[2], but developing effective medicines targeting these receptors has proven challenging, partly due to an incomplete molecular level understanding of how receptors are engaged and activated by chemokines, and what governs their coupling to the principal intracellular effectors, G proteins and arrestins[3,4].

Advances in cryo-EM have led to a significant increase in the number of experimentally determined receptor-chemokine complexes (23 chemokine complex cryo-EM structures of 12 receptors in the PDB as of December 2024[5], out of the total of 63 structures of 15 chemokine receptors by all methods) and has shed light on molecular activation mechanisms. A general two-site paradigm has been established in which the chemokine globular core provides binding affinity and specificity while its N-terminus drives activation[6–12]. However, alternative mechanisms have been suggested[13–18], and further study of receptor complexes will be necessary to refine the basis of both ligand engagement on the extracellular face and effector coupling on the intracellular face of chemokine receptors. Artificial-intelligence-powered computational modeling tools have the potential to complement experimental structure determination in this endeavor. For example, the breakthrough AlphaFold2 (AF2) technology[19,20] often delivers GPCR-peptide complex models with accuracy comparable to cryo-EM structures[21]. Moreover, in certain cases it can generate conformational ensembles[22–24], with the structural variation across the ensemble resembling the motions observed in molecular dynamics (MD) simulations[24].

C-C chemokine receptor 9 (CCR9) is a chemokine receptor that is expressed on subsets of developing thymocytes and intestinal

[1]Department of Pathology and Immunology, Faculty of Medicine, University of Geneva, Geneva, Switzerland. [2]Skaggs School of Pharmacy and Pharmaceutical Sciences, University of California San Diego, La Jolla, CA, USA. [3]Orion Biotechnology, Campus Biotech Innovation Park, Geneva, Switzerland. [4]These authors contributed equally: Inês De Magalhaes Pinheiro, John R. D. Dawson, Kalyana Bharati Akondi. ✉e-mail: ikufareva@ucsd.edu; oliver.hartley@unige.ch

lymphocytes[25]. Through interaction with its only chemokine ligand, CCL25, CCR9 promotes migration of these cells into their target organs (the thymus and small intestine, respectively) in the context of immune maintenance, surveillance and inflammation[26,27]. The CCR9-CCL25 axis has attracted interest as a target for the treatment of inflammatory bowel disease[28], and has been studied in the context of tumor progression[29] and immuno-oncology[30]. However, no CCR9-targeting therapeutics have received regulatory approval. While an inactive structure of CCR9 in complex with the allosteric inhibitor vercirnon has been determined[31], the molecular mechanisms of CCR9 agonism remain unknown.

Here, we map the determinants of CCR9 activation by CCL25 using AF2 structural modeling and pharmacological assessment of rationally selected CCR9 binding pocket mutants. We show that for CCR9, the main driver of activation is a structure located in the globular core of CCL25 – the 30s loop. These findings challenge the established two-site model for chemokine receptor engagement[6–12], highlight the diversity of molecular mechanisms underlying chemokine receptor activation, and inform rational structure-based targeting of these receptors.

## Results

### Structural features of the CCR9-CCL25 signaling complex

To gain insight into the interaction of CCR9 with CCL25, we constructed a series of AlphaFold2 and AlphaFold3 models of the CCR9-CCL25 signaling complex (Fig. 1A, Supplementary Data 1–3). The complex features the overall architecture of all canonical receptor-chemokine complexes revealed by experimental structure determination to-date: the proximal N-terminus of the receptor (chemokine recognition site 1 or CRS1) binds in the surface groove between the N-loop and 40s loop of the chemokine core, the chemokine N-terminus is submerged in the orthosteric binding pocket (CRS2),

and the areas surrounding the conserved disulfides of the chemokine and the receptor pack against each other and form the intermediate CRS1.5 (Fig. 1A; here and elsewhere, interactive ICM Browser sessions[32] for the molecular figures are provided in Supplementary Data 4). Most parts of the complex were predicted with high confidence (pLDDT scores > 90) and minimal conformational variability across the model ensemble (Supplementary Figs. 1–3); however, lower prediction confidence (Supplementary Fig. 1) and increased variability (Supplementary Fig. 4) were observed for the distal N-terminus of the chemokine (pLDDT scores of 45–57.5, 50–56, and 58–75 for the chemokine N-terminal residues 1, 2, and 3, respectively).

Superposition of the highest confidence AF2 model onto the X-ray structure of inactive CCR9 bound to vercirnon[31] revealed that CCL25-bound CCR9 features an outward movement of the intracellular ends of the transmembrane (TM) helices 5 and 6 (Fig. 1B), a shared activation signature across the GPCR superfamily. Prominent TM helix rearrangements are also apparent in the orthosteric binding pocket on the extracellular side of the receptor (Fig. 1C): compared to inactive CCR9, the CCL25-bound conformation features a large inward movement of the extracellular end of TM5 with a concurrent outward movement of extracellular loop (ECL) 3 and the adjoining ends of TM6 and TM7, providing an opening to accommodate the chemokine. These movements are consistent with activation-associated rearrangements on the extracellular side of other chemokine receptors[33], but not necessarily characteristic of other GPCR subfamilies. Finally, and uniquely among the chemokine receptors studied so far, CCR9 activation is also accompanied by a profound (~6 Å) downward (i.e. in the intracellular direction) sliding-and-bending motion of TM5, and, to a lesser extent of TM6, relative to the rest of the TM bundle (Fig. 1D).

To better understand the shared and unique architectural features of the CCR9-CCL25 complex, we compared it to the available experimental structures of other active-state receptor-chemokine

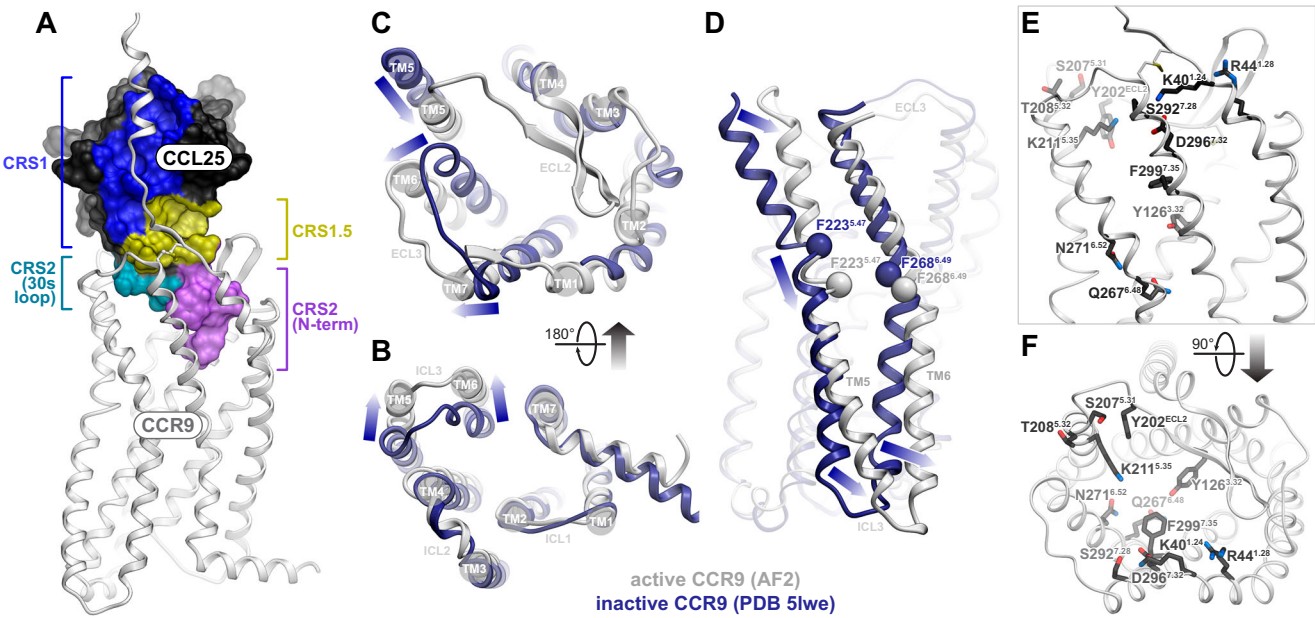

**Fig. 1 | Structural model of CCR9 signaling complex with CCL25 and rationale for receptor mutagenesis. A** AlphaFold2 model of the CCR9-CCL25 signaling complex, viewed along the plane of the membrane. The receptor is shown as a white ribbon, the chemokine as a black mesh where surfaces interacting with the indicated receptor regions are colored. CRS: chemokine recognition site. Model coordinates are available in Supplementary Data 1, interactive ICM Browser[32] sessions for this and other figures in Supplementary Data 4. **B**–**D** Structural superposition of CCR9 in its predicted CCL25-bound conformation (white) with the X-ray structure of CCR9 bound to the small molecule antagonist vercirnon (PDB entry

5LWE, navy) viewed perpendicular to the plane of the membrane from the intracellular or extracellular side (**B** and **C**, respectively), or parallel to the plane of the membrane in the TM5-to-TM2 direction (**D**). Arrows indicate the directions of the structural changes between the inactive and the predicted active states. In **D**, the Cα atoms of reference residues close to the middle of TM5 and TM6 are shown as spheres. **E**, **F** CCR9 CRS2 residue positions selected for mutagenesis are shown as sticks and viewed parallel to the membrane (**E**) or perpendicular to the membrane from the extracellular side (**F**).

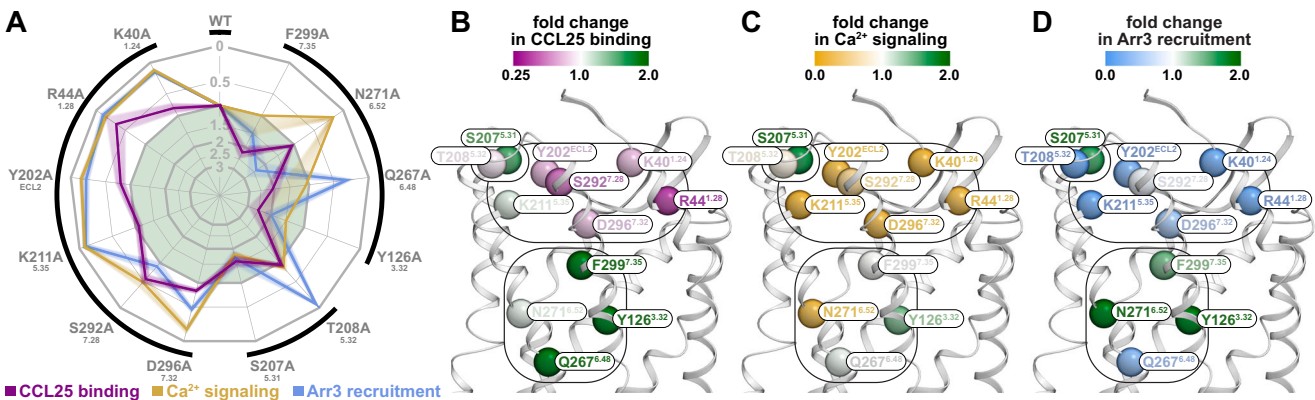

**Fig. 2 | Overview of the impact of CCR9 mutations on CCL25 binding and CCL25-induced signaling. A** Radar plot summarizing the impact of CCR9 mutations on CCL25 binding (purple), CCL25-induced intracellular $Ca^{2+}$ mobilization (orange), and CCL25-induced Arr3 recruitment to the receptor (blue). Mutation impacts are expressed as ratios of mutant to WT responses in respective experiments; contours outside or inside the central light green area correspond to negative and positive impacts, respectively. Responses were determined as areas under the CCL25 concentration-response curves (AUCRCs) for $Ca^{2+}$ mobilization and Arr3 recruitment, and as receptor-specific cell fluorescence increases in the presence of 300 nM of TAMRA-labeled CCL25 for binding. $Ca^{2+}$ mobilization and Arr3 recruitment data represent mean values from independent experiments (*n* = 3). Binding data represent mean values from independent experiments (*n* = 5 for parental and

WT CCR9; *n* = 3 for K40A, R44A, K211A, D296A, S207A, T208A, Y126A; *n* = 2 for Y202A, S292A, Q267A, N271A, F299A). SEM values, used for descriptive purposes only, are represented by the contour width and transparency at the respective mutant axis. Black external brackets denote the groups of functionally and structurally related mutations as they are presented in this paper. Statistics are available in Supplementary Table 1. Detrimental or beneficial impact of mutations at selected residues is reflected in the color of their Cα atoms (spheres), in relation to CCL25 binding (**B**), CCL25-induced intracellular $Ca^{2+}$ mobilization (**C**), and CCL25-induced Arr3 recruitment to CCR9 (**D**). The receptor is shown as a white ribbon and viewed parallel to the plane of the membrane. Rounded rectangles mark the groups of functionally and structurally related mutations, matching the outside brackets in (**A**).

complexes[14,17,34–38]. CCR9 belongs to the same phylogenetic subfamily as CCR1, CCR2, CCR5 and CCR6, sharing 40-45% identity in the TM domains (CCR6 is the closest with 44.9% TM identity) and 30-35% identity in the TM regions and extracellular loops involved in chemokine binding (CCR5 is the closest with 34.7% identity). At the primary sequence level, the CCR6 ligand CCL20 is one of the most similar to CCL25, sharing the characteristics of a short N-terminal region (six residues in CCL25 versus five in CCL20; other chemokines have seven or more) and an unusually long 30s loop (11 residues in CCL25, 29-QEVSGSCNLPA-39 – versus 9 in CCL20; other chemokines have 7 or less, Supplementary Fig. 5A). Despite these sequence similarities, the predicted binding mode of CCL25 to CCR9 is strikingly different from the experimentally determined CCR6-CCL20 complex[34] and more closely resembles the interactions of CCR1, CCR2 and CCR5 with their respective chemokines (Supplementary Fig. 5B-G): whereas the 30s loop of CCL20 lies above the binding pocket of CCR6 (Supplementary Fig. 5E), the CCL25 30s loop enters the CCR9 binding pocket together with the N-terminal region and forms prominent interactions with CCR9 CRS2 (Fig. 1A).

### Evaluation of the roles of CCR9 CRS2 residues in chemokine binding and signaling

To probe the functional significance of the observed CRS2 interactions between CCL25 and CCR9, we selected 10 residues in CCR9 CRS2 (TM domains only) whose sidechains make strongest direct contacts with the chemokine in multiple models of the AF2 ensemble (Fig. 1E, F, Supplementary Fig. 6) and generated alanine substitution mutants. We added two further alanine mutants at Ballesteros-Weinstein (BW)[39] positions 6.48 and 6.52 (CCR9 Q267[6.48] and N271[6.52]). Residues at these positions do not directly contact CCL25 in the models but are known to play critical roles in extracellular-to-intracellular signal transmission in other chemokine and non-chemokine GPCRs[40–43]. The 12 mutants were characterized (Supplementary Fig. 7) to determine the relative impact of each position on CCL25 binding and signaling. Binding studies (Supplementary Fig. 7A) were performed at four different concentrations of CCL25 site-specifically labeled with rhodamine at its C-terminal extremity (Supplementary Fig. 8). To determine relative impact in

binding we made use of data points from the highest concentration (300 nM), which provided the strongest discrimination between the CCR9-dependent binding signal and signal from CCR9-independent binding to cell surface proteoglycans (Supplementary Fig. 8). For signaling studies, we distinguished between G protein- and arrestin-driven activities using a calcium flux assay (downstream of G protein activation, Supplementary Fig. 7B) and a BRET-based arrestin3 recruitment assay (Supplementary Fig. 7C). Both assays were carried out at six different CCL25 concentrations, with the area under the concentration-response curve used to quantify signaling activity and assess the relative impact of mutations.

The mutations had a broad range of effects, ranging from complete abrogation of binding or signaling to more than 2-fold enhancement compared to WT, as evidenced by the non-circular shapes of the assay contours in Fig. 2A. The binding and the two signaling readouts were not always affected in a consistent manner as reflected by divergent assay contour shapes (Fig. 2A): some mutants negatively impacted all three experimental readouts (e.g. R44[1.28]A, Fig. 2A), others impacted signaling to a greater extent than binding (e.g. K211[5.35]A, Fig. 2A), and some selectively abrogated one signaling response ($Ca^{2+}$ mobilization or Arr3 recruitment) while making no difference in binding or the other signaling readout (e.g. T208[5.32]A and N271[6.52]A, Fig. 2A). When viewed on the receptor model, the mutations framing the opening of the pocket generally decreased all three readouts, whereas those located on the periphery or deeper in the binding pocket affected the readouts in divergent ways (Fig. 2B–D). Based on spatial location and impact, we tentatively grouped the mutations as shown in Fig. 2B-D and investigated the specific molecular interactions that are responsible for the observed alterations in chemokine binding and receptor signaling by mutations in each group.

### The extracellular "rim" of CCR9 helical bundle is critical for engaging CCL25 in a signaling-productive conformation

Across the entire panel, four mutations stood out as being practically signaling-dead (less than 15% of WT signaling output): K40[1.24]A, R44[1.28]A, Y202[ECL2]A, and K211[5.35]A (Fig. 3A-D, Supplementary Figs. 9 and 10). For R44[1.28]A, the loss of signaling could be attributed to

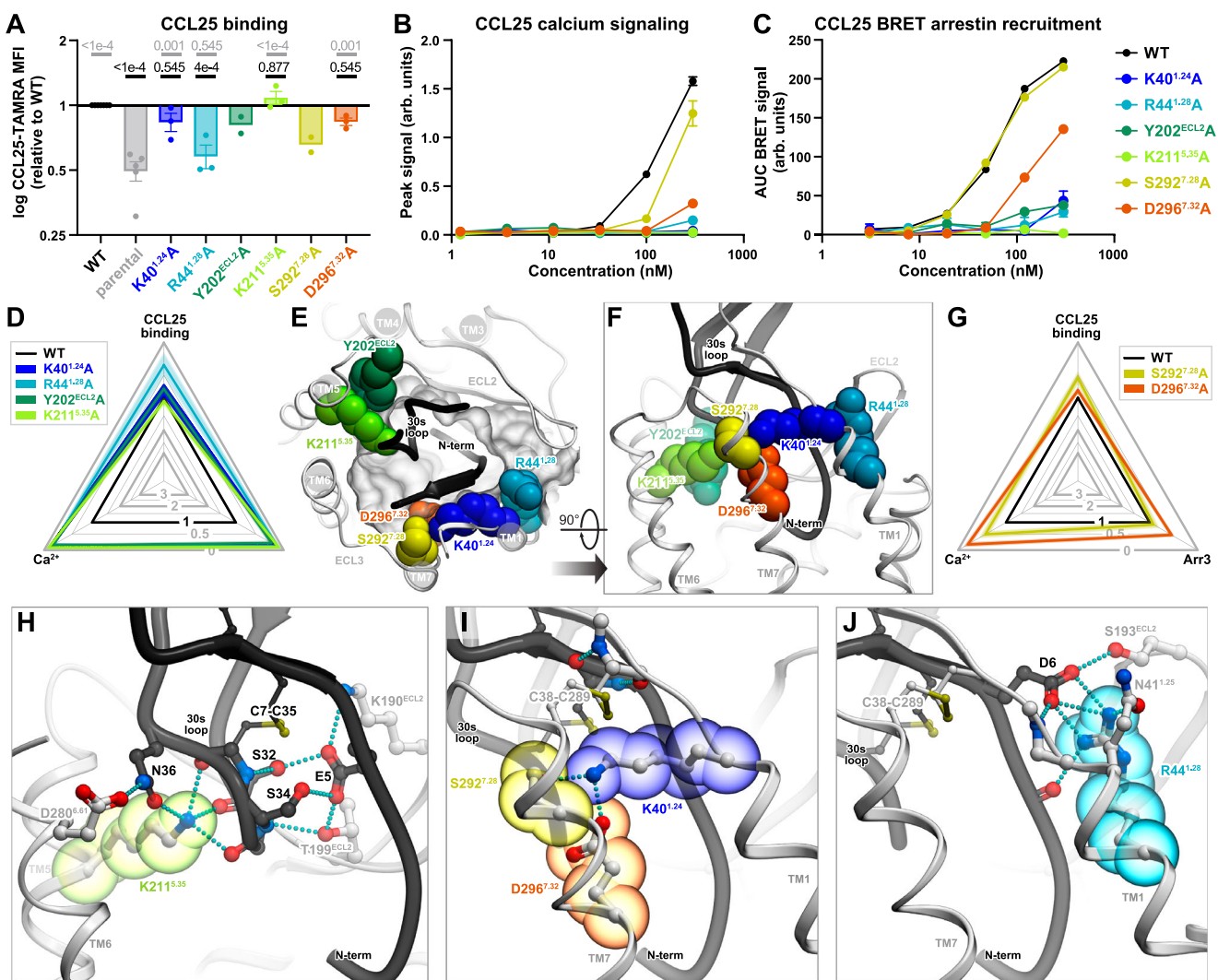

**Fig. 3 | Pharmacological and structural evaluation of the mutations in the extracellular "rim" of the CCR9 helical bundle.** Statistical analyses are shown in Supplementary Table 1. Source data are provided in **Source Data** files. **A** Binding of 300 nM TAMRA-CCL25 to WT and mutant CCR9 in HEK293T cells. Bars show mean ± SEM binding ratios (mutant:WT) from independent experiments ($n$ = 5 for WT and parental, $n$ = 3 for K40A, R44A, K211A, D296A, and $n$ = 2 for Y202A and S292A). Where applicable ($n$ > 2), $P$-values (one-way ANOVA on log-ratios with post-hoc tests and Holm-Šídák's correction for multiple comparisons) versus WT and parental cells are shown in black and grey, respectively. Complete CCL25 binding CRCs in Supplementary Fig. 9. **B** CCL25 concentration responses of WT and mutant CCR9 in the Ca²⁺ mobilization assay. Shown are mean peak signals ± SEM from triplicate wells in a single experiment, representative of 3 independent experiments (Supplementary Fig. 10A–C). **C** CCL25 concentration responses of WT and mutant CCR9 in the BRET-based Arr3 recruitment assay. Shown are means ± SEM of BRET signals obtained in triplicate wells in a single experiment, representative of 3

independent experiments (Supplementary Fig. 10D-F). **D**, **G** Radar plot data from Fig. 2A showing the impact of the indicated mutants on CCL25 binding and signaling. The black triangle denotes WT CCR9; for mutants, scalene contours[113] indicate disproportionate mutation impact across assays. **E**, **F** The indicated residues (colored spheres) form part of a "rim" of the CCR9 (white ribbon) helical bundle and interact with the chemokine proximal N-terminus and 30s loop (black ribbon). **E** extracellular view across membrane plane; binding pocket is shown as a transparent mesh; chemokine globular core is clipped for clarity. **F** view along the membrane plane. **H–J** Hydrogen-bonding networks (cyan dotted lines) of CCR9 K211⁵·³⁵ (**H**), K40¹·²⁴ (**I**), and R44¹·²⁸ (**J**). The complex is viewed along the plane of the membrane, color scheme is the same as in **E**, **F**. WT: wild-type. TAMRA: tetra-methylrhodamine, MFI: median fluorescence intensity, ns: not significant, BRET: bioluminescence resonance energy transfer, AUC: area under curve, TM: transmembrane, ECL: extracellular loop.

a complete loss of binding (Fig. 3A, the mutant is indistinguishable from the negative control, the parental cell line). However, for the remaining three mutants, the binding was fully or partially preserved (Fig. 3A).

In the CCR9-CCL25 model, K40¹·²⁴, R44¹·²⁸, Y202^ECL2, and K211⁵·³⁵ are part of the extracellular rim of the TM bundle that 'grips' the chemokine to hold it in the binding pocket (Fig. 3E). K40¹·²⁴ and R44¹·²⁸ primarily coordinate the proximal N-terminus of CCL25 while Y202^ECL2 and K211⁵·³⁵ act on the 30s loop (Fig. 3E). Proximal to K40¹·²⁴ are TM7 residues S292⁷·²⁸ and D296⁷·³² that also contribute to the 'grip' (Fig. 3E, F). Functionally, the D296⁷·³²A mutation nearly eliminated the Ca²⁺

response and greatly reduced arrestin recruitment (only 15% and 46% of WT output retained, respectively), with no loss of binding, whereas S292⁷·²⁸A significantly weakened the binding and Ca²⁺ response, but retained arrestin recruitment (48% and 89%, respectively, Fig. 3A-C, G, Supplementary Figs. 9 and 10).

Closer examination of the predicted CCR9-CCL25 complex reveals that these residues are centers of three critical hydrogen bonding networks. First, K211⁵·³⁵ forms three hydrogen bonds with the backbones of the CCL25 30s loop residues S32, G33 and C35, and in addition bridges the chemokine residue N36 to D280⁶·⁶¹, thus affixing the 30s loop to CCR9 TMs 5 and 6 (Fig. 3H). Second, K40¹·²⁴ shapes the

junction between CCR9 TMs 1 and 7 by hydrogen-bonding to both S292[7.28] and D296[7.32]; this also improves packing of its side chain with the proximal N-terminus of CCL25 (Fig. 3I). Third, R44[1.28] is involved in an intricate network connecting the proximal N-terminus of the CCL25 (including residue D6, conserved in a subset of chemokines, Supplementary Fig. 5A, and important for their signaling[34,44]), CCR9 TM1 (including the backbone of K40[1.24]), and CCR9 ECL2 (S193[ECL2]) (Fig. 3J). Completing the assembly, the proximal N-terminus of the chemokine and its 30s loop are connected not only through the C7-C35 disulfide bridge but also by hydrogen bonding between CCL25 S32, E5, and S34, with E5 also hydrogen-bonding to K190[ECL2] and T199[45.51] in CCR9 ECL2 (Fig. 3H). Y202[ECL2] provides steric interactions and structural support to the ECL2-30s loop-TM5 side of the assembly (Fig. 3E).

Altogether, these results delineate the role of the extracellular "rim" of the CCR9 orthosteric pocket in binding and positioning the proximal N-terminus and the 30s loop of the chemokine in a signaling-productive conformation. Importantly, disrupting the hub of interactions involving K211[5.35] and Y202[ECL2] had a disproportionate effect on signaling relative to binding (Fig. 3A–D); this suggests that the 30s loop of CCL25, including residue N36, drives CCR9 signaling through TM5 and TM6. The involvement of CCR9 TM5 and TM6 is consistent with the rearrangements of their extracellular ends observed in the active-inactive structure comparison (Fig. 1C, D). The central signaling role of the 30s loop is unusual in the chemokine receptor family[18] and may be a unique feature of the CCR9-CCL25 complex.

### Top-of-TM5 mutations can enhance and bias CCR9 signaling in response to CCL25

Two additional residues were mutated at the periphery of the CCR9 TM domain: S207[5.31] and T208[5.32]. Despite being solvent-facing and adjacent to each other, these residues produced pronounced and strikingly distinct effects when mutated to alanine. S207[5.31]A strongly enhanced CCL25 binding (Fig. 4A and Supplementary Fig. 11A–C) together with CCL25-induced $Ca^{2+}$ flux (Fig. 4B and Supplementary Fig. 12A–C) and Arr3 recruitment (Fig. 4C, D and Supplementary Fig. 12D–F). By contrast, T208[5.32]A completely abrogated CCL25-mediated Arr3 association (Fig. 4C, D and Supplementary Fig. 12D–F) while leaving CCL25 binding (Fig. 4A and Supplementary Fig. 11A–C) and $Ca^{2+}$ signaling (Fig. 4B and Supplementary Fig. 12A–C) unchanged. The full G protein signaling capacity of CCR9 T208[5.32]A was confirmed in a non-amplified G protein activation assay (Gαi/Gβγ dissociation BRET, Supplementary Fig. 13).

In the complex model, S207[5.31] is positioned one helical turn above K211[5.35] and is proximal to an entirely hydrophobic surface on CCL25 30's loop (Fig. 4E-G). The elimination of S207[5.31] hydroxyl group via an alanine mutation would strengthen CCR9 hydrophobic packing against this surface, which explains the observed concerted increase in chemokine binding and agonism. T208[5.32] interacts with an adjacent part of the chemokine surface that features only a single polar atom, the backbone oxygen of residue N36. Interestingly, in the top-ranked (by pLDDT[19]) model of the complex, the rotamer state of T208[5.32] and its distance from N36 were not conducive to the formation of a hydrogen bond. However, a hydrogen-bond forming conformation was identified when examining the entire ensemble[22,24] of AF2-generated CCR9-CCL25 models (Fig. 4H, Supplementary Data 1, Supplementary Data 4). Moreover, this conformation corresponds to what we interpret as the "most active" state of CCR9, based on three features emphasized by the active-inactive structure comparison in Fig. 1B-D: the largest outward movement of the intracellular end of TM6, the greatest intramolecular distance across the binding pocket in the TM3-to-TM6/7 direction, and the "deepest" position of TM5 relative to the rest of the TM helices (Fig. 4H–J). We hypothesize that Arr3 recruitment requires this "most active" state involving the T208[5.32]-N36 hydrogen bond, whereas G protein association is permissive to a range of active-like conformations of CCR9, as previously described for other

GPCRs[45]. The loss of T208[5.32]-N36 hydrogen bonding in the T208[5.32]A mutant would therefore selectively abrogate Arr3 recruitment with minimal impact on G protein-mediated $Ca^{2+}$ mobilization.

Altogether, these results establish distinct and nontrivial roles for two extracellularly facing, partially solvent-exposed residues S207[5.31] and T208[5.32] in controlling not only chemokine binding but also receptor signaling and bias. Worth noting is the proximity of these residues to K211[5.35] and the chemokine 30s loop, consistent with the role of TM5 as a driver of CCR9 activation and the 30s loop as a major signaling determinant in the CCR9-CCL25 complex.

### Non-canonical roles of residues at and below the binding pocket floor in CCR9 activation

Next, we turned our attention to the residues deeper in the binding pocket - Y126[3.32] and F299[7.35], - and in the middle of the TM domain of CCR9 - Q267[6.48] and N271[6.52].

Y126[3.32] is located at the "floor" of the binding pocket, is highly conserved across the chemokine receptor family, and has been shown to be a critical signal initiation determinant as its mutations abrogate signaling in many receptors[43,46–60]. Surprisingly, the CCR9 Y126[3.32]A mutation did not have a negative impact; instead, it led to a striking increase (2.5-fold) in CCL25 binding (Fig. 5A, D and Supplementary Fig. 14) accompanied by modest increases in $Ca^{2+}$ signaling and arrestin recruitment (Fig. 5B–D and Supplementary Fig. 15). A 2.2-fold increase in CCL25 binding was also observed for the F299[7.35]A mutant (Fig. 5A, D and Supplementary Fig. 14), with no effects on signaling (Fig. 5B, D and Supplementary Fig. 15). In the model, Y126[3.32] is in direct contact with the N-terminal pyroglutamate (pGlu1) of CCL25 (expected to form from Q1 in the mature chemokine) while F299[7.35] participates in a perpendicular/T-shaped stacking with CCL25 residue F4 (Fig. 5E, F). The observation that these two residues can be eliminated without negatively impacting CCL25 signaling suggests that, in contrast to other receptor-chemokine pairs[18], interactions between the distal N-terminus of CCL25 and the floor of CCR9 do not play a major role in either receptor binding or activation.

The remaining two residues, Q267[6.48] and N271[6.52], belong to TM6 and are positioned below the pocket "floor" (Fig. 5F). Q267[6.48] corresponds to the "toggle-switch" residue and is a tryptophan in most GPCRs. Its substitution by alanine substantially increased CCL25 binding and fully preserved $Ca^{2+}$ signaling, while reducing Arr3 recruitment to 29% of the WT signal (Fig. 5A-D, Supplementary Figs. 11, 12); the full G protein signaling capacity of this mutant was corroborated in a non-amplified G protein subunit dissociation assay (Supplementary Fig. 16). N271[6.52]A had no effect on CCL25 binding, but had an opposite effect compared to Q267[6.48], enhancing Arr3 recruitment and almost completely abrogating $Ca^{2+}$ signaling (15% of WT response remaining, Fig. 5A, D, Supplementary Figs. 14, 15).

The observed enhancement of chemokine binding and, in some mutants, signaling through one or both pathways could not be explained by improved intermolecular contacts: according to the structural model, the Y126[3.32]A and F299[7.35]A mutations eliminate contacts with the chemokine, and for residues Q267[6.48] and N271[6.52], direct chemokine contact is not possible at all. We therefore considered an alternative explanation in which the mutations promote chemokine binding and signaling by shifting the CCR9 conformational equilibrium towards the active, chemokine-compatible state. Indeed, in the inactive CCR9 structure[31], residues Y126[3.32], Q267[6.48], and N271[6.52] are proximal to each other and form numerous steric and polar contacts (Fig. 5G); their separation is exclusive to the active state of CCR9 (Fig. 5F), is concurrent with the outward movement of TM6 and TM7 (Fig. 1C), and thus can serve as a marker of CCR9 activation, as suggested by Fig. 1B–D and Fig. 4. By disrupting the Y126[3.32]-Q267[6.48]-N271[6.52] interaction cluster (Fig. 5G) and eliminating the inactive-state-specific cross-pocket coordination, alanine mutations of the participating residues are likely to destabilize the inactive state of CCR9 and

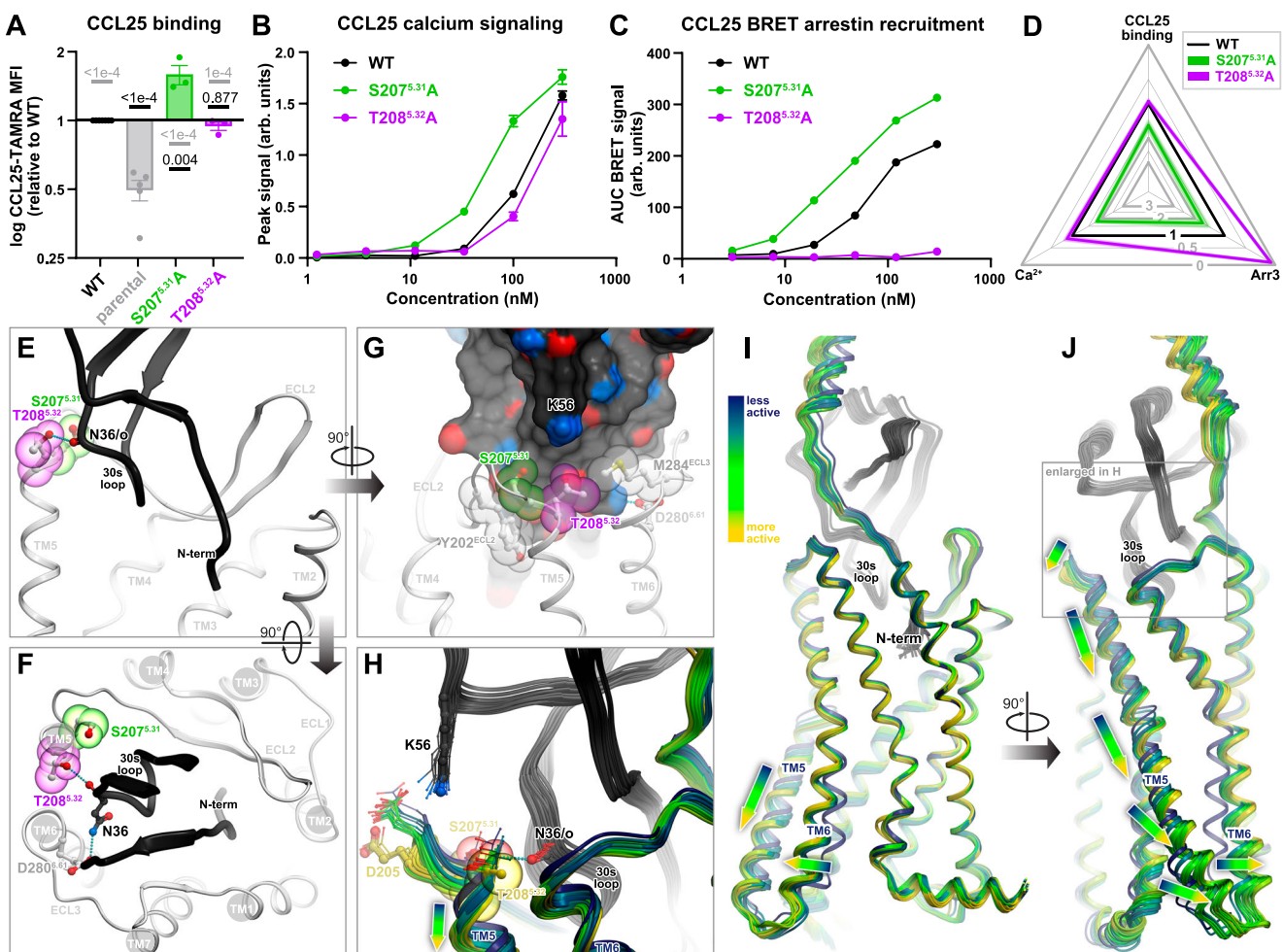

**Fig. 4 | Pharmacological and structural evaluation of the peripheral top-of-TM5 CCR9 mutations.** Statistical analyses are shown in Supplementary Table 1. Source data are provided in **Source Data** files. **A** CCL25 binding to WT and mutant CCR9 in HEK293T cells. Bars show mean ± SEM mutant:WT binding ratios from independent experiments ($n$ = 5 for parental and WT CCR9, $n$ = 3 for S207A and T208A). $P$-values (one-way ANOVA with post-hoc tests and Holm-Šídák's correction) versus WT and parental cells are shown in black and grey, respectively. Complete CCL25 binding CRCs in Supplementary Fig. 11. **B** CCL25 concentration responses of WT and mutant CCR9 in the Ca²⁺ signaling assay. Mean peak signals ± SEM from triplicate wells in a single experiment, representative of 3 independent experiments (Supplementary Fig. 12A–C). **C** CCL25 concentration responses of WT and mutant CCR9 in the Arr3 recruitment assay. Mean BRET signals ± SEM from triplicate wells in a single experiment, representative of 3 independent experiments (Supplementary Fig. 12D-F). **D** Radar plot data from Fig. 2A showing mutation impact on CCL25 binding and signaling. Black triangle denotes WT CCR9; scalene contours[113] indicate disproportionate mutation impact across assays. **E, F** Location of CCR9 S207⁵·³¹ and T208⁵·³² (sticks and colored spheres) in relation to CCL25 (black ribbon) in the complex. Hydrogen bonds are shown as cyan dotted lines. **E** view along membrane plane with receptor TM helices 1, 6, and 7 removed for clarity. **F** extracellular view across membrane with CCL25 globular core clipped for clarity. **G** Packing of CCR9 S207⁵·³¹ and T208⁵·³² against CCL25 30s loop. The CCL25 surface mesh shows exposed hydrogen bond donors (red), acceptors (blue), and nonpolar atoms (black). View along membrane plane in TM5-to-TM2 direction. **H–J** Variations of the "degree of receptor activation" in the CCR9-CCL25 AF2 ensemble. CCR9 activation (color rainbow) is quantified as the distance between Y126³·³² hydroxyl and N271⁶·⁵² side-chain amide (Fig. 1B-D). Arrows show direction of conformational changes as Y126-N271 distance increases. **H** thicker sticks and spheres denote the "most active" model (gold) with an H-bond to CCL25 N36 shown as a cyan dotted line. Ensemble coordinates in Supplementary Data 1. WT: wild-type. TAMRA tetramethylrhodamine, MFI median fluorescence intensity, BRET bioluminescence resonance energy transfer, AUC area under curve, TM transmembrane, ECL extracellular loop.

make the active state more prevalent even in the absence of chemokine, i.e. introduce receptor constitutive activity. Consistent with this, all four mutants showed significantly increased association with Arr3 in the absence of chemokine, as indicated by elevated basal BRET in the HEK-CCR9-RLuc8 cell lines (Supplementary Fig. 17). Assessment of relative surface (by flow cytometry) vs total (by luminometry) receptor levels (Supplementary Fig. 18D) suggests that Y126³·³²A and F299⁷·³⁵A, but not Q267⁶·⁴⁸A or N271⁶·⁵²A, are predominantly intracellular, indicative of constitutive internalization: a common feature of constitutively active, Arr3-associated receptors[61].

Beyond predicting this constitutive activity, the computational models were unable to explain the striking signaling bias of the Q267⁶·⁴⁸A and N271⁶·⁵²A mutants. However, the single helical turn that separates

Q267⁶·⁴⁸ and N271⁶·⁵² in the CCR9 structure harbors P269⁶·⁵⁰, the most conserved amino acid in TM6 of class A GPCRs and the core of the TM6 kink. Moreover, mutations of Q267⁶·⁴⁸ and N271⁶·⁵² selectively eliminate TM6 contacts with TM7 and TM5, respectively (Supplementary Fig. 19). Therefore, we hypothesize that the loss of these contacts alters the conformational coupling between the binding site and the intracellular effector interface in a manner that preferentially affects Arr3 or G protein.

Collectively, these results indicate that, in contrast to many other chemokine receptors, the residues at or below the floor of the CCR9 binding pocket are not the determinants of chemokine binding. However, they appear to subtly control the conformational preferences of the receptor, not only in the active-inactive spectrum but also in effector selectivity.

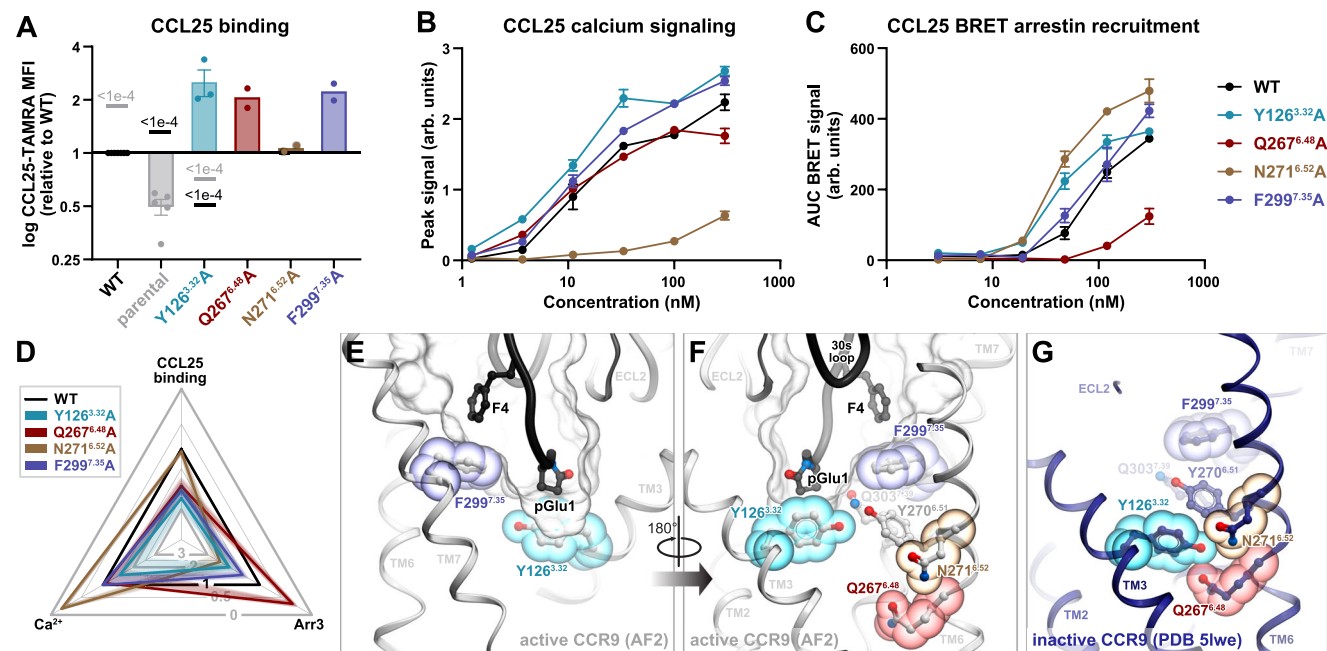

**Fig. 5 | Pharmacological and structural evaluation of the CCR9 mutations at and below the binding pocket floor.** Statistical analyses are shown in Supplementary Table 1. Source data are provided in **Source Data** files. **A** CCL25 binding to WT and mutant CCR9 in HEK293T cells. Bars show mean ± SEM binding ratios (mutant vs WT) from independent experiments ($n = 5$ for parental and WT CCR9, $n = 3$ for Y126A, and $n = 2$ for Q276A, N271A, F299A). Where applicable ($n > 2$), $P$-values (one-way ANOVA with post-hoc tests and Holm-Šídák's correction on log ratios) versus WT and parental cells are shown in black and grey, respectively. Complete CCL25 binding CRCs in Supplementary Fig. 14. **B** CCL25 concentration responses of WT and mutant CCR9 in the $Ca^{2+}$ signaling assay. Shown are mean peak signals ± SEM from triplicate wells in a single representative experiment out of 3 (Supplementary Fig. 15A–C). **C** CCL25 concentration responses of WT and mutant CCR9 in the Arr3 recruitment assay. Mean ± SEM of BRET signals from triplicate wells in a single representative experiment of 3 (Supplementary Fig. 15D-F). **D** Radar plot data from Fig. 2A showing impact of indicated mutants on CCL25 binding and signaling. Black triangle denotes WT CCR9; scalene contours[113] indicate disproportionate mutation impact across assays. **E** Location of CCR9 Y126[3.32] and F299[7.35] (sticks and colored spheres) relative to the orthosteric pocket (transparent mesh) and the N-terminus of the chemokine (black ribbon and sticks) in the complex. View along the membrane plane in the TM2-to-TM5 direction with TMs 1 and 2 omitted for clarity. **F–G** Location of CCR9 Q267[6.48] and N271[6.52] (sticks, dark-red and brown spheres) relative to the orthosteric pocket (mesh), the chemokine (black ribbon), and Y126[3.32]/F299[7.35] in the predicted complex (**F**) or the X-ray structure of inactive CCR9[7] (**G**). Views along the membrane plane in the TM5-to-TM2 direction with TMs 4 and 5 omitted for clarity. WT wild-type, TAMRA tetramethylrhodamine, MFI median fluorescence intensity, BRET bioluminescence resonance energy transfer, AUC area under curve, AF2 AlphaFold2, PDB Protein Data Bank, TM transmembrane, ECL extracellular loop.

## Partial alanine scanning of CCL25 confirms the key role of its 30s loop in CCR9 activation

To further probe the key sites of interaction in the active CCR9-CCL25 complex, we generated a series of CCL25 alanine mutants targeting the N-terminal region (positions 1-6) and the 30s loop (positions 28-37). These mutants were assessed for their capacity to activate CCR9 using a $Ca^{2+}$ flux assay on the MOLT-4 human T leukemia cell line, which endogenously expresses CCR9[62] (Fig. 6A–D, Supplementary Fig. 20).

Within the N-terminal region of CCL25, only mutation of D6 led to a decrease in signaling; G2A and V3A enhanced signaling and all other mutations had no impact (Fig. 6A, D). The impact of the D6A mutation was very mild, contrasting e.g. its role in CCL20[34,44]. In contrast, seven out of nine alanine mutations in the 30s loop reduced signaling, while only G33A enhanced signaling and E30A had no effect. The mutations only affected signaling potency ($EC_{50}$, Fig. 6D) but not efficacy ($E_{max}$, Fig. 6A–C), suggesting that CCR9 binding affinity was modulated but the ability to promote full agonism was preserved.

According to the structural model, CCL25 D6 and N36 are integral parts of the hydrogen bond networks surrounding CCR9 R44[1.28] (Fig. 3J) and K211[5.35] (Fig. 3I), respectively, while CCL25 V31 provides important steric packing interactions for CCR9 Y202[ECL2], S207[5.31], T208[5.32], and K211[5.35] (Fig. 6E-G). CCL25 S32 forms an intramolecular hydrogen bond network with CCL25 S34 and CCL25 E5 (Fig. 6G), likely contributing to the stabilization of CCL25 in a signaling-productive conformation. The aliphatic sidechain of L37 contributes to intramolecular packing and is also in contact with CCR9 residue M284[ECL3] (Fig. 6F), which is proximal to CCL25 N36 and CCR9 T208[5.32] and

D280[6.61] (Fig. 4G). This suggests a role for L37 in stabilizing the TM5/6-30s loop interaction. The side chain of Q29 is buried in the chemokine core, forming two hydrogen bonds with the backbone of the proximal N-terminus C8 and one additional bond with the backbone of L37, thus stabilizing the 30s loop shape (Fig. 6F). The CCL25 30s loop residues with lower mutation impact also form prominent intramolecular interactions: the aliphatic side chain of I28 packs against the surrounding chemokine residues (Fig. 6F) and S34 hydrogen-bonds to CCL25 E5 (Fig. 6G), and together with S32 they likely stabilize CCL25 in a signaling-productive conformation.

These results provide further evidence that it is the 30s loop rather than the N-terminus of CCL25 that carries the critical determinants of CCR9 activation by forming key hydrogen bonding networks and hydrophobic interactions with residues in CCR9 TM5 and TM6.

## Discovery of a 'super-binder' CCL25 analog [1P6]CCL25

To further understand the structural regions that drive the capacity of CCL25 to bind and activate CCR9, we used a previously described phage display-based chemokine engineering strategy[63]. From libraries with diversity introduced into various N-terminal positions (Supplementary Table 3 and Supplementary Fig. 21) we identified [1P6]CCL25, an analog with enhanced receptor engagement capacity, that contains N-terminal residues Y1-Q2-A3-S4 in place of pGlu1-G2-V3-F4 (Fig. 7A). Compared to WT CCL25, [1P6]CCL25 showed a substantial increase in CCR9 binding (Fig. 7B and Supplementary Fig. 22A–C), similarly to previously reported phage-display-generated analogs of CCL5[64–66] and CXCL12[67]. [1P6]CCL25 also demonstrated enhanced CCR9 signaling

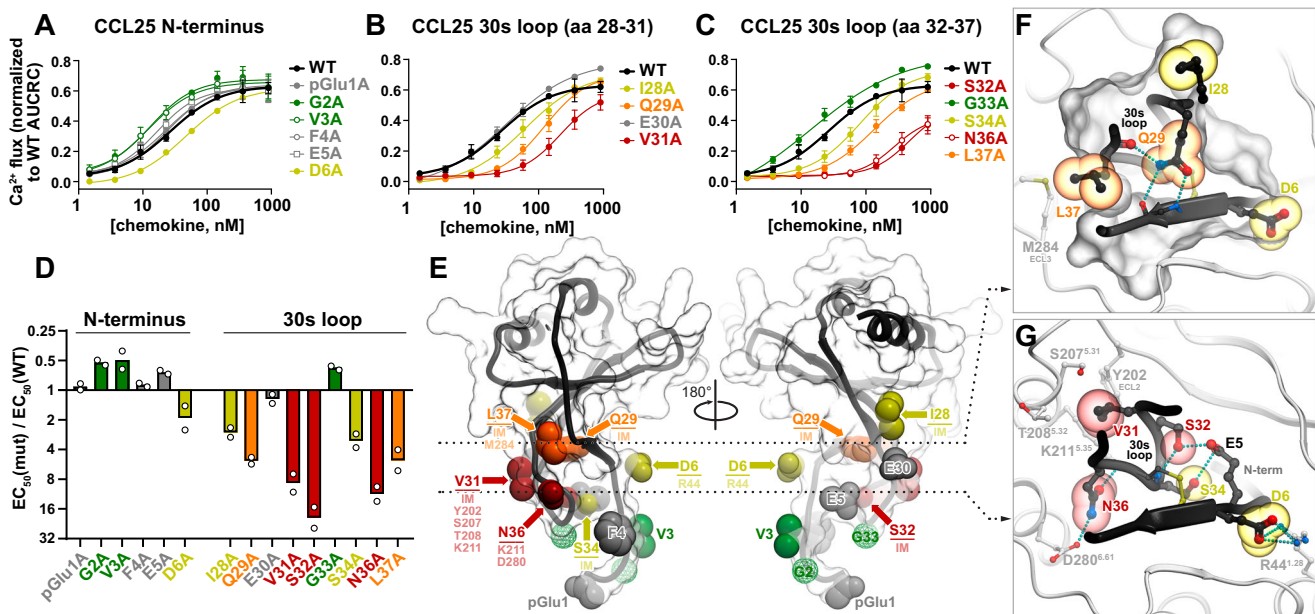

**Fig. 6 | Pharmacological evaluation of CCL25 N-terminus and 30s loop mutants.** Statistical analyses are shown in Supplementary Table 2. Source data are provided in **Source Data** files. **A–C** Concentration-dependent Ca²⁺ signaling responses of MOLT-4 cells to WT CCL25 and the indicated CCL25 mutants. Data points represent mean normalized peak signals at the indicated ligand concentrations from 2 independent experiments (Supplementary Fig. 20). **D** Bar graph summarizing the data in **A–C**. Shown are mean $EC_{50}$ ratios (mutant:WT CCL25) from 2 independent experiments. Bars are colored red-to-green based on the mutation impact on chemokine signaling potency; bars pointing up and down represent stronger and weaker mutant potency, respectively, compared to WT CCL25. **E** Mutation impacts projected on the 3D model of CCL25 (black ribbon and a transparent mesh) from the CCR9-CCL25 complex. For mutated residues, the atoms eliminated (or, for glycine residues, introduced) via an alanine mutation are shown as spheres and colored as in **A–D**. For each residue with a measurable negative impact, its major intra- and intermolecular interactions are labeled. IM: intramolecular interaction. **F**, **G** Focused view of the intra- and intermolecular interactions involving the chemokine residues whose mutations have a negative impact on CCR9 signaling: D6, I28, Q29, and L37 (**F**) and V31, S32, S34, and N36 (**G**). The chemokine is shown in black ribbon and sticks (and a surface mesh in **F**), mutated residues in spheres colored as in **A–E**. The receptor is shown in white ribbon and sticks. The complex is viewed across the plane of the membrane from the extracellular side; **F** and **G** correspond to the cross-sectional planes indicated by dashed lines in **E**. Hydrogen bonds are shown as cyan dotted lines.

activity, both in terms of Ca²⁺ mobilization and Arr3 recruitment (Fig. 7C, D and Supplementary Fig. 22D–I).

The predicted binding mode of [1P6]CCL25 to CCR9 is similar to that of WT CCL25, with the residues shared between the two ligands making the same interactions with the receptor (Fig. 7E, F, Supplementary Data 1, Supplementary Data 4, 5). However, notable differences are apparent in both the conformation of the distal N-terminus and the receptor interactions of the four substituted residues (Fig. 7G, H).

To provide a structural explanation for the enhanced binding and signaling activity of [1P6]CCL25, we used RTCNN[33,68,69], an AI-based scoring function for protein-ligand interactions. RTCNN scores of chemokine atoms were aggregated to residue backbones and side chains to estimate their respective contributions to the complex binding affinity. This suggested that the most favorable CCL25 N-terminus contributions are from residues E5 and D6 (the latter engaged with the CCR9 extracellular "rim" amino acid R44[1.28], Fig. 3J), which are shared by the two ligands (Fig. 7G, H). Per RTCNN, only two of the four N-terminal amino-acids, pGlu1 and F4, contribute favorably to binding of WT CCL25: pGlu1 packs against Y126[3.32] and W104[2.60], whereas F4 perpendicularly stacks with F299[7.35]. The roles of the remaining two WT CCL25 N-terminal residues, G2 and V3, are predicted to be neutral or minimally advantageous (Fig. 7G), in agreement with alanine mutagenesis results (Fig. 6D). In contrast, all four of the substituted N-terminal amino acids in [1P6]CCL25 contribute favorably to CCR9 binding. First, the positively charged N-terminal amine of Y1 (unavailable in WT CCL25 due to pGlu being cyclized) forms a cation-pi interaction with W104[2.60] and hydrogen-bonds to Y55[1.55] and Q303[7.39], while its side chain packs against TM1 (A47[1.31] and L51[1.35]) and TM2 (A107[2.63] and A108[2.64]) (Fig. 7H). Second, the side chain of Q2 is oriented towards the 30s loop and stabilizes it in a beneficial conformation through hydrogen bonding. Third, mutating V3

in WT CCL25 to A3 in [1P6]CCL25 reduces bulk to accommodate the Y1 sidechain. Finally, the side chain of [1P6]CCL25 S4 forms a hydrogen bond with D296[7.32].

These results demonstrate that engineering the N-terminal region of CCL25 can generate ligands with enhanced binding and signaling via the introduction of beneficial interactions with CRS2 of CCR9. However, in contrast with N-terminal molecular evolution studies of other chemokines[64,67], our study did not yield receptor antagonists (Supplementary Table 3, Supplementary Fig. 21), suggesting that CCR9 activation determinants are located outside of the N-terminal domain of CCL25, likely in the 30s loop as suggested by the data in Figs. 3–6.

## Super-binder [1P6]CCL25 is more tolerant to CCR9 mutations than WT CCL25

Next, we investigated the impact of the 12 CCR9 CRS2 point mutations (Fig. 1E, F) on the binding and signaling of the super-binder [1P6]CCL25 compared to WT CCL25. The mutations affected the two ligands in a broadly similar way, with the direction of impact (positive or negative) generally preserved for each mutation. However, ligand-specific differences in the magnitudes of the changes for certain mutations were observed (Fig. 8). In contrast to their varying negative effects on CCL25 binding (ranging from complete abrogation to no impact, Fig. 3), the six extracellular 'rim' mutations uniformly but only partially decreased [1P6]CCL25 binding (Fig. 8A and Supplementary Fig. 23). For the remaining mutations, binding was either unaffected or enhanced for both ligands, with the relative enhancements less pronounced for [1P6]CCL25 (Fig. 8A). For Ca²⁺ mobilization and Arr3 recruitment, the positive and negative mutation impacts observed for WT CCL25 were either preserved or attenuated for [1P6]CCL25 (Fig. 8B, C, E, F and Supplementary Fig. 23).

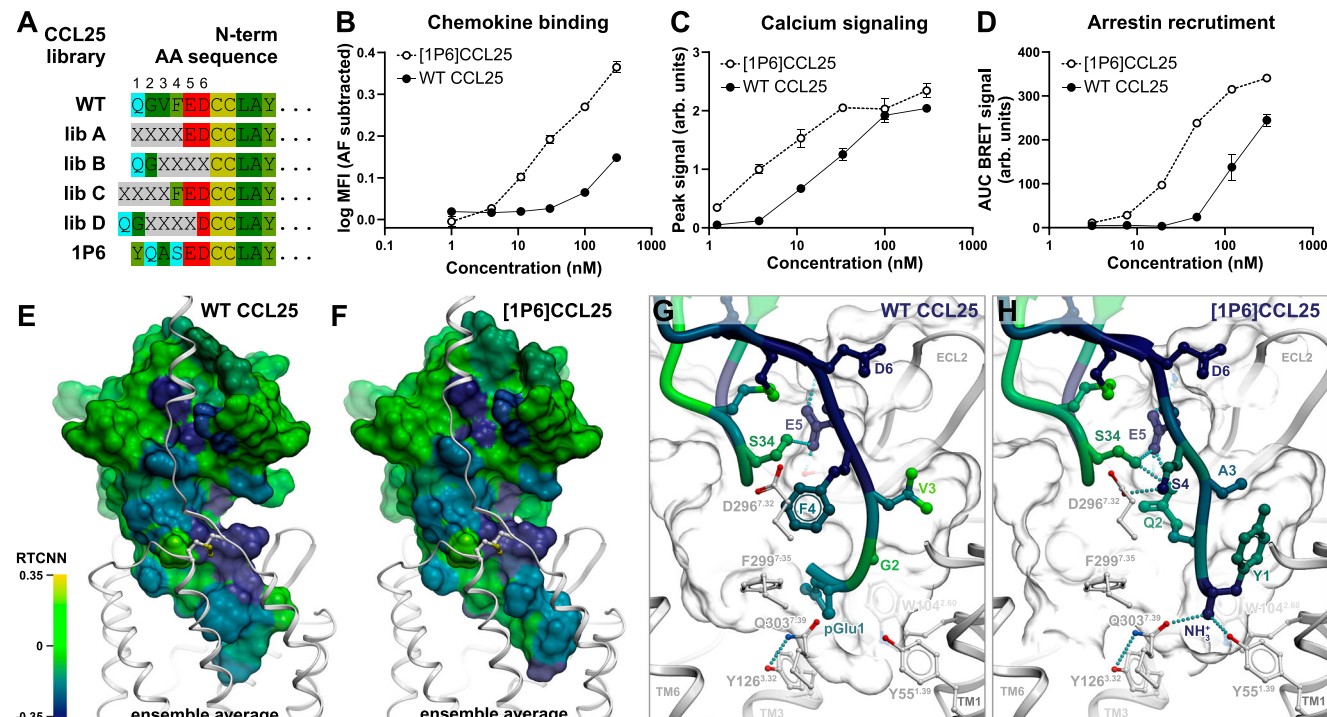

**Fig. 7 | [1P6]CCL25 is an N-terminally engineered super-binder and super-agonist analog of CCL25.** Source data are provided as **Source Data** files. **A** CCL25 phage libraries used in this study, indicating N-terminal sequences of native CCL25 and the analog [1P6]CCL25. X = fully randomized residue; residues from Cys7 correspond to CCL25(7-73). N-terminal Q is expected to be cyclized to pGlu in the mature protein. Analog sequences are available in Supplementary Table 3. **B** Concentration-dependent binding of TAMRA-labeled CCL25 or [1P6]CCL25 to CCR9 stably expressed in HEK293T cells. Data represent mean ± SEM of specific binding signal from triplicate wells in a single representative experiment out of 3 independent experiments (Supplementary Fig. 23A, D, G). **C** Concentration-dependent CCR9 responses to CCL25 and [1P6]CCL25 in the Ca²⁺ signaling assay. Data represent mean peak Ca²⁺ signals ± SEM in response to the indicated ligand concentrations in triplicate wells in a single experiment, representative of 3 independent experiments (Supplementary Fig. 23B, E, H). **D** Concentration-dependent

Arr3 recruitment to CCR9 by CCL25 and [1P6]CCL25. Data represent mean ± SEM of BRET signals from triplicate wells in a single experiment and are representative of 3 independent experiments (Supplementary Fig. 23C, F, I). **E, F** Top-scoring predicted conformations of CCL25 (**E**) and [1P6]CCL25 (**F**) (rainbow-colored surfaces) bound to CCR9 (white ribbons). Chemokines are colored by per-atom RTCNN scores averaged across the top 5 models from each model ensemble and aggregated to residue backbones and side chains. The complexes are viewed along the membrane plane. **G, H** Zoomed-in views of the chemokine N-termini (ribbons and sticks) in the top-scoring models of CCR9-CCL25 (**G**) and CCR9-[1P6]CCL25 (**H**) complexes. Color represents per-atom RTCNN scores aggregated to residue backbones (for ribbons and backbone sticks) and side chains (for side chain sticks). Ribbons for receptor TM helices 7 and (partially) 1 are omitted for clarity. Model coordinates are available in Supplementary Data 1.

Collectively, these results suggest that the improved binding properties of [1P6]CCL25, mediated by the enhanced interactions of its distal N-terminus with CCR9 CRS2, not only make it a more potent agonist of the receptor but also more tolerant to CRS2 mutations.

## Discussion

This study presents a comprehensive map of the interaction interface between CCR9 and its endogenous agonist CCL25, with delineation of determinants of binding, signaling, constitutive activity, and bias. A key feature of our structural model is the depth to which the 30s loop of CCL25 accompanies the N-terminus into the TM domain of CCR9 (Fig. 1A). In this respect, the CCR9-CCL25 complex resembles the experimentally determined structures of CCR1, CCR2, and CCR5 with their respective chemokine agonists (Supplementary Figs. 5B-G), but differs from those of CCR6, CXCR2, and CX3CR1, in which the 30s loops of the bound chemokines do not engage the TM domain, but instead interact with the extracellular loops[15,70] (Supplementary Figs. 5E–G). An extensive hydrogen bond network linking the 30s loop and the proximal N-terminus acts together with the first conserved disulfide bridge to fuse the two chemokine regions into a single structural unit.

Functional mapping of the CCR9-CCL25 interface revealed a noncanonical role for the CCL25 N-terminus. For many chemokines, the N-terminus serves as a 'message' for receptor activation[6–12];

however, in the case of CCL25 and CCR9, it appears to make a minimal contribution to either binding or signaling. Mutating receptor residues predicted to contact the N-terminus, or alanine substitution of CCL25 N-terminal residues, did not have any negative impact on chemokine binding or receptor activation (Figs. 5, 6). However, increased binding could be achieved through molecular evolution of the CCL25 N-terminus, as illustrated by the potent chemokine analog [1P6]CCL25 (Fig. 7). Importantly, such molecular evolution did not produce changes in chemokine signaling unattributable to alterations in binding. This provides further evidence that the N-terminus of CCL25 is not the driver of signaling, and contrasts with findings for many other chemokines, where N-terminal modifications strongly affect signaling properties[8,60,64,67,71–73]. Previous studies have identified some deviations from the two-site model, such as the tolerance of CCR6-CCL20[34,44] or the atypical chemokine receptor, ACKR3[67,74,75], to modifications at the distal N-terminus of the chemokine. However, these studies still highlighted residues at the proximal N-terminus as critical contributors to receptor activation.

In support of a non-canonical signaling anatomy of the CCR9-CCL25 complex, we showed that the 30s loop of CCL25, rather than its N-terminus, is the principal determinant of receptor activation. The loop engages in extensive interactions with TM5, the domain that undergoes profound lateral and longitudinal motions upon receptor activation (Fig. 1C, D). Mutations of CCR9 TM5 and ECL2 residues that

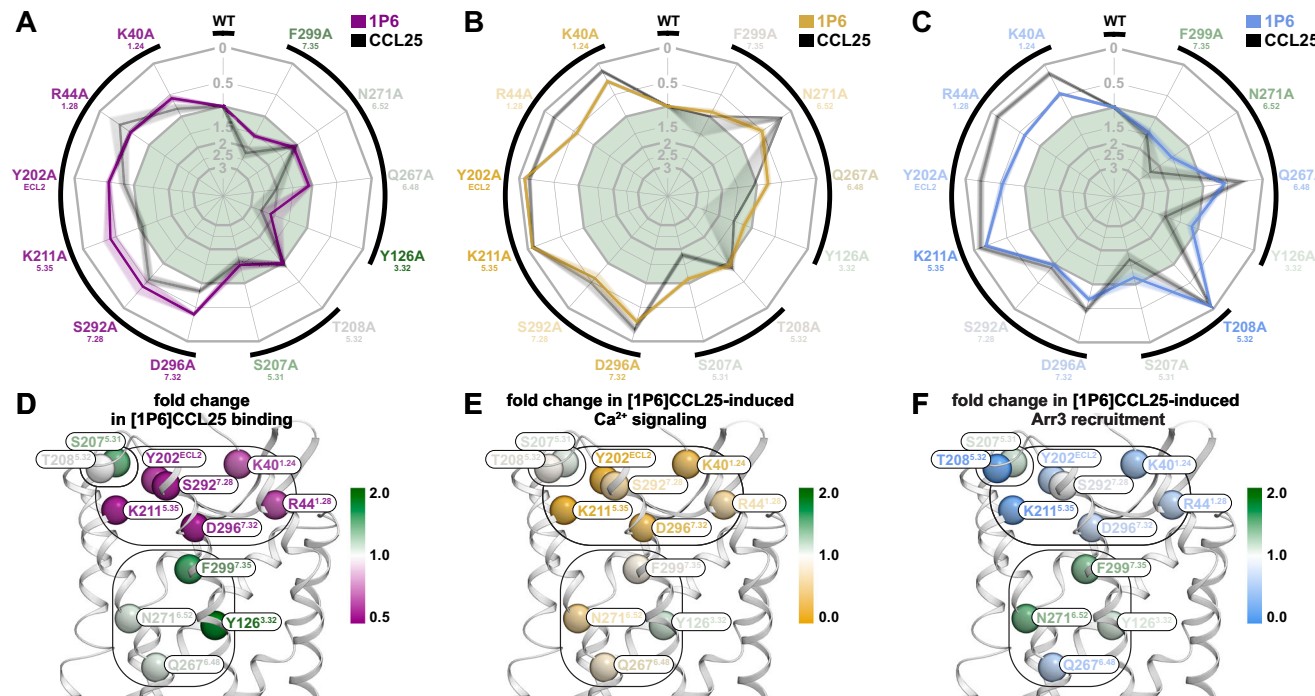

**Fig. 8 | [1P6]CCL25 has increased tolerance to CRS2 mutations in CCR9.**
**A**–**C** Radar plots summarizing the impact of studied CCR9 mutations on [1P6]
CCL25 binding (purple), [1P6]CCL25-induced intracellular Ca²⁺ mobilization
(orange), and [1P6]CCL25-induced Arr3 recruitment to the receptor (blue). The
impact of the same mutations on the binding and signaling of WT CCL25 (data from
Fig. 2) is shown in grayscale for reference. Mutation impacts are expressed as ratios
of CCR9 mutant to WT responses in respective experiments; contours outside or
inside the central light green area correspond to negative and positive impacts,
respectively. Responses were determined as areas under the [1P6]CCL25
concentration-response curves (AUCRCs) for Ca²⁺ mobilization and Arr3 recruit-
ment, and as receptor-specific cell fluorescence increases in the presence of
300 nM of TAMRA-labeled [1P6]CCL25 for binding. Data represent mean of inde-
pendent experiments (n = 3 for WT CCR9 and for K40A, R44A, K211A, S207A,

T208A, Y126A, and D296A; n = 2 for Y202A, S292A, Q267A, N271A, and F299A); SEM
values, shown for descriptive purposes only, are represented by the contour width
and transparency at the respective mutant axis. Black outside brackets denote the
groups of functionally and structurally related mutations presented in Figs. 3, 4,
and 5. Summary data and all individual replicates are shown from Supplementary
Fig. 24 to Supplementary Fig. 29. Statistics are available in Supplementary Table 4.
**D**–**F** Detrimental or beneficial impact of mutations at selected residues is reflected
in the color of their Cα atoms (spheres), in relation to [1P6]CCL25 binding (**D**), [1P6]
CCL25-induced intracellular Ca²⁺ mobilization (**E**), and [1P6]CCL25-inducedArr3
recruitment to CCR9 (**F**). The receptor is shown as white ribbons and viewed par-
allel to the plane of the membrane. Rounded rectangles mark the groups of func-
tionally and structurally related mutations presented in this paper.

are in contact with the 30s loop abrogated signaling (K211⁵·³⁵ and
Y202ᴱᶜᴸ²) or made it strongly biased (T208⁵·³²), with no effect on che-
mokine binding. However, the 30s loop of CCL25 also plays a key role
in receptor binding: the majority of 30s loop mutations led to sig-
nificant decreases in Ca²⁺ flux potency but not efficacy, consistent with
loss of receptor binding affinity.

Interpretation of GPCR structure-function studies is invariably
complicated by the potential for individual receptor mutants to dif-
ferentially impact aspects of receptor function such as constitutive
activity, G protein and arrestin-coupling preferences, intracellular
trafficking and signaling from subcellular compartments[64,76–80]. In our
study, we addressed this challenge by assessing the impact of receptor
and chemokine mutations on multiple aspects of receptor pharma-
cology. By systematically and quantitatively evaluating CCR9 mutation
impacts on chemokine binding, calcium mobilization, and Arr3
recruitment, we identified prominent examples of Ca²⁺-biased
(T208⁵·³²A and Q267⁶·⁴⁸A) and Arr3-biased (N271⁶·⁵²A) mutants. Con-
scious of the potential amplification artefacts inherent to Ca²⁺ flux and
other second messenger assays[60,81–83] we confirmed full G protein
competence of mutants (T208⁵·³²A and Q267⁶·⁴⁸A) in a non-amplified
BRET-based Gαi/Gβγ dissociation assay. By measuring the basal
arrestin association (Supplementary Fig. 17), we identified mutants
with high levels of constitutive activity (Y126³·³²A, F299⁷·³⁵A, Q267⁶·⁴⁸A
and N271⁶·⁵²A), and, by assessing surface and total receptor expression
(Supplementary Fig. 18A-D), inferred altered subcellular distribution
for two of them (Y126³·³²A and F299⁷·³⁵A). Finally, all data was

contextualized in an ensemble of structural models of the CCR9-CCL25
complex, allowing us to deconvolute molecular mechanisms under-
lying the altered pharmacology of the mutants.

The structural ensembles[22,24] were made possible through the use
of AF2. Although the final and ultimate validation of predicted inter-
molecular contacts would require experimental structure determina-
tion or at least charge-swap[84] or disulfide crosslinking[85] mutagenesis,
AF2 has been shown to readily generate near-experimental accuracy
models for GPCR-peptide complexes[21]. Expanding on the previously
solved X-ray structure of inactive, antagonist-bound CCR9 ([31], Fig. 1B-
D), AF2 model ensemble provided a plausible basis for partial vs full
activation of CCR9 (Fig. 4H–J) and the requirements for Arr3 recruit-
ment (Fig. 4). Nonetheless, the ability of AF2 to predict structural
dynamics is limited[23,86–88]. Accordingly, certain conformational states
of the complex remained inaccessible to our modeling, as did the
entropic component of binding. As a consequence, we were only able
to provide tentative explanations for the biased activity of three
receptor mutants (T208⁵·³²A, Q267⁶·⁴⁸A and N271⁶·⁵²A) and the affinity
improvements of selected chemokine variants (G2A, V3A, and G33A).

Arrestin-versus-G protein signaling bias is a therapeutically
exploitable phenomenon[89,90] that has also been observed between the
natural ligands of several members of the chemokine receptor family[91].
Unfortunately, despite recent progress, structural understanding of
such bias remains elusive. Available for only three GPCRs so far,
NTR1[92], CNR1[93], and OPSD[94], complementary experimental structures
with G proteins and with arrestins have not revealed a generalizable

conformational signature for signaling bias. Accordingly, future studies of the CCR9-CCL25 interaction using molecular dynamics[74,95], NMR[96], single-molecule FRET[97,98], or time-resolved cryo-EM[99] may provide a more detailed explanation of the signaling biases of the mutants identified in our work.

The impact of our study extends beyond understanding the structure-function relationships in the chemokine receptor family. Polymorphisms that alter CCR9 residues studied here have been identified in the human population and include Y126$^{3.32}$H, S207$^{5.31}$N, T208$^{5.32}$I, Q267$^{6.48}$H, N271$^{6.52}$D/S, and F299$^{7.35}$L[100,101]. However, any functional consequences these polymorphisms might engender have yet to be described. By revealing the roles of affected residues in shaping the CCR9 response to the endogenous chemokine agonist, our study provides a rationale for possible alterations of CCR9-mediated immune responses in variant carriers and paves the way towards personalized medicine[102].

In summary, this study reveals that CCR9-CCL25 is a receptor-chemokine pair with non-canonical mechanisms of engagement and signaling, adding diversity to the established two-site model of chemokine receptor activation. Our results suggest that engineering of the 30s loop of CCL25 can yield potent CCR9 modulators with tunable signaling activity and add to the growing portfolio of chemokine analogs suitable for clinical development[103]. They also provide insights for structure-based design of small molecule therapeutics for CCR9-related pathologies.

## Methods

### Model building

The model building workflow is schematically represented in Supplementary Fig. 30. Structural models of CCR9A complexes with WT CCL25 and [1P6]CCL25 were built by AlphaFold2 Multimer v2.3.2[19,20,104] locally installed on the UCSD Triton Shared Computing Cluster (TSCC). The amino-acid sequences used were:

```
CCR9:
MTPTDFTSPI PNMADDYGSE STSSMEDYVN FNFTDFYCEK NNVRQFASHF
LPPLYWLVFI
VGALGNSLVI LVYWYCTRVK TMTDMFLLNL AIADLLFLVT LPFWAIAAAD
QWKFQTFMCK
VVNSMYKMNF YSCVLLIMCI SVDRYIAIAQ AMRAHTWREK RLLYSKMVCF
TIWVLAAALC
IPEILYSQIK EESGIAICTM VYPSDESTKL KSAVLTLKVI LGFFLPFVVM
ACCYTIIIHT
LIQAKKSSKH KALKVTITVL TVFVLSQFPY NCILLVQTID AYAMFISNCA
VSTNIDICFQ
VTQTIAFFHS CLNPVLYVFV GERFRRDLVK TLKNLGCISQ AQWVSFTRRE
GSLKLSSMLL
ETTSGALSL

CCL25:
PGVFEDCCLA YHYPIGWAVL RRAWTYRIQE VSGSCNLPAA IFYLPKRHRK
VCGNPKSREV
QRAMKLLLDAR NKVFAKLHHN TQTFQAGPHA VKKLSSGNSK LSSSKFSNPI
SSSKRNVSLL
ISANSGL

[1P6]CCL25:
YQASEDCCLA YHYPIGWAVL RRAWTYRIQE VSGSCNLPAA IFYLPKRHRK
VCGNPKSREV
QRAMKLLLDAR NKVFAKLHHN TQTFQAGPHA VKKLSSGNSK LSSSKFSNPI
SSSKRNVSLL
ISANSGL
```

Initially, for each complex, an ensemble of 25 models (5 seeds and 5 models per seed) was built. Using ICM software version 3.9-3b[105] the WT CCL25 and [1P6]CCL25 models were modified to include a cyclized pGlu1 and a free positively charged N-terminus (NH$_3^+$), respectively. Chemokine molecules in all complexes were subjected to local gradient minimization with positional harmonic restraints on Cα atoms using ICM[105]. Complexes were then scored using the Radial and Topological Convolutional Neural Network (RTCNN), a deep learning-based scoring function implemented in ICM and trained to distinguish protein complexes with potent binders from similar decoy complexes[33,68,69]. High-scoring complexes were prioritized in the analysis.

For the best-scoring CCR9-[1P6]CCL25 model, additional refinement was performed in ICM in two stages: one employing 3D grid potentials and another full-atom representation of all components. During the first stage, the receptor binding pocket was represented with a set of grid interaction potentials, including those for van der Waals, electrostatic, hydrogen bonding, and apolar surface interactions[106,107]. The N-terminus and the 30s loop of [1P6]CCL25 (Tyr1-QASEDC-Cys8 and Ile28-QEVSGSCNL-Pro38 respectively) were built ab initio; an explicit disulfide bond was imposed between C7 and C35; and residues C8, I28, and P38 were tethered to the corresponding residues in the template. The conformational stack of the system, including the N-terminus and 30s loop was initialized based on the AlphaFold2 model ensemble, and the system was then thoroughly sampled in the receptor potential grids, using biased probability Monte Carlo sampling in ICM, to optimize and expand on the conformational stack. For the second stage, the resulting conformational stack was merged with full atom models of the receptor, and at least 10$^8$ steps of Monte Carlo optimization were performed, allowing for the same level of flexibility in the chemokine fragments with added full flexibility of receptor binding pocket sidechains. For full-atom sampling, van der Waals, torsional, hydrogen bonding, electrostatic, and disulfide bond energy terms were used. The resulting conformations were clustered, re-scored using RTCNN, inspected visually, and the binding geometry of the best-scoring fragmented complex was transferred onto the full [1P6]CCL25 model for visualization.

For in-depth assessment of prediction confidence and conformational variability, an additional 500 models (100 seeds with 5 models per seed) were built for full-length CCR9 with WT CCL25(1-85) using AF2 on TSCC (Supplementary Data 2). Additional 100 models (20 seeds with 5 models per seed) were built for each of the CCR9-CCL25(1-85) (Supplementary Data 3) and CCR9-[1P6]CCL25(1-85) (Supplementary Data 5) complexes using the AlphaFold server implementation of AF3[108]. These models are featured in Supplementary Figs. 2-4.

### Reagents

All commercially available reagents used in this study are listed in Supplementary Table 5.

### Chemokines

Chemokines were prepared by Fmoc solid phase peptide synthesis[109]. Following cleavage, peptides were ether-precipitated and folded using a glutathione redox buffer (2 m guanidine hydrochloride, 0.1 M Tris base, 0.5 mm reduced GSH, 0.3 mM oxidized GSH, and 10 mM methionine (pH 8.0). Purity and integrity of products were routinely verified by analytical reversed-phase (RP-) HPLC and mass spectrometry.

The chemokines used in this study are based on a previously described C-terminally truncated version of CCL25 (CCL25(1-73)[110]). For preparations of CCL25 and [1P6]CCL25, Met$^{64}$ was substituted for norleucine, an azidolysine residue was appended to the C-terminal end, and chemokines were subjected to a column purification step

(RP-HPLC) prior to and following the folding reaction. All other chemokine samples were produced using a previously described column-free method[109]. CCL25 samples prepared using the column purified and column-free methods were shown to have indistinguishable signaling activity from each other and from that of full-length (1-127) recombinant CCL25 (Supplementary Fig. 31). Fluorescent versions of CCL25 and [1P6]CCL25 were generated by coupling an excess of tetramethylrhodamine (TAMRA)-PEG4-DBCO to the azido-lysine lateral chain. Excess dye was removed by 10 kDa size exclusion purification.

### Plasmids
Previously generated plasmids used in this study are listed in Supplementary Table 6. For the newly generated FUGW lentiviral vectors encoding CCR9 and CCR9 C-terminally fused to the *Renilla* luciferase variant RLuc8 (CCR9-RLuc8), and variants carrying single alanine mutations (K40A, R44A, Y126A, Y202A, S207A, T208A, K211A, Q267A, N271A, S292A, D296A and F299A), genes were synthesized and subcloned by GenScript using previously described template plasmids.

### Cell culture
HEK293T parental cells and HEK293T cells expressing CCR9 (HEK-CCR9) and CCR9 Ala mutants were cultured in Dulbecco's modified Eagle's medium (DMEM) supplemented with 10% fetal bovine serum (FBS) and 1% Penicillin-Streptomycin. MOLT-4 cells and CHO cells stably expressing CCR9 (CHO-CCR9) were cultured in RPMI-1640 supplemented with 10% FBS and 1% Penicillin-Streptomycin. Cells were grown in a humidified incubator at 37 °C with 5% $CO_2$.

### Chemokine phage display
Chemokine phage display was performed as described in ref. 63. Four phage libraries of CCL25 variants were generated, each incorporating full randomization of four residues in the N-terminus, with two libraries featuring a one-residue N-terminal extension. The four phage libraries were combined and subjected to selection on CHO-CCR9 cells. 48 enriched sequences identified after the third and fourth rounds of selection were chosen for further evaluation.

### Generation of CCR9 expressing cell lines
HEK-CCR9 and HEK-CCR9-RLuc8 YFP-arrestin 3 (Arr3) cells, and respective alanine mutants, were obtained by lentiviral transduction as previously described[111], followed by selection of high-expressing populations via fluorescence-activated cell sorting (FACS) in a BD FACS Aria® Fusion flow cytometer using a fluorescent anti-CCR9 monoclonal antibody (anti-CCR9 mAb; BD Biosciences). For arrestin recruitment reporter cell lines, HEK293T parental cells were first lentivirally transduced with FUGW-YFP-Arr3, and a high-YFP-expressing population was selected by FACS. The selected FUGW-YFP-Arr3 cell population was then lentivirally transduced with appropriate FUGW-CCR9-RLuc8 vectors, followed by population selection by FACS with anti-CCR9 mAb (Gating strategy summarized in Supplementary Fig. 32).

### Receptor surface level quantification via flow cytometry
Over the course of the work, CCR9 expression in HEK-CCR9 and HEK-CCR9-RLuc8 YFP-Arr3 cell lines was regularly measured using flow cytometry with a fluorescent anti-CCR9 mAb (Supplementary Fig. 18A, B) (Gating strategy summarized in Supplementary Fig. 32). For this, following detachment with 0.5 mM EDTA, $2 \times 10^5$ cells per sample were incubated with an anti-CCR9 mAb (1:100 dilution) in FACS buffer (1x PBS, 1 mM EDTA, 1% BSA) in 96-well plates. Following 1 h incubation, cells were washed once in FACS buffer and mAb binding was measured by flow cytometry on a Cytoflex instrument (Beckman Coulter). $10^4$ events were collected per sample, in technical triplicates, and median fluorescence intensity (MFI) values of anti-CCR9 mAb were obtained using CytExpert (Beckman Coulter). Antibody binding signals were expressed as:

$$\frac{\log MFI_{mut} - \log MFI_{parental}}{\log MFI_{WT} - \log MFI_{parental}} \tag{1}$$

The antibody is directed against an unknown epitope on CCR9; to exclude the possibility of its binding being affected by individual Ala mutations, we assessed correlation of antibody binding between the untagged CCR9 mutants and their respective CCR9-RLuc8 counterparts. A mutation affecting antibody binding would be expected to alter surface detection levels for both receptor mutant variants; however, no correlation was found (Supplementary Fig. 18C). Total expression in HEK-CCR9-RLuc8 cell lines was assessed by luminometry and compared with the surface expression in the same cell lines (Supplementary Fig. 18D).

### Flow cytometry-based chemokine binding assays
Following detachment with 0.5 mM EDTA, $2 \times 10^5$ cells per sample were incubated TAMRA-labeled chemokines diluted in FACS buffer (1x PBS, 1 mM EDTA, 1% BSA) in 96-well plates. Following 1 h incubation, cells were washed once in FACS buffer and ligand binding was measured by flow cytometry on a Cytoflex instrument (Beckman Coulter) (Gating strategy summarized in Supplementary Fig. 32). $10^4$ events were collected per sample, in technical triplicates, and median fluorescence intensity (MFI) values of CCL25-TAMRA and [1P6]CCL25-TAMRA were obtained using CytExpert (Beckman Coulter). Nonspecific binding was measured in parental HEK293T cells (Supplementary Fig. 8). For binding evaluation of CCR9 mutants, we used the highest concentration (300 nM) of fluorescent chemokine, and binding signals were expressed as:

$$\frac{\log MFI_{mut} - \log AF_{mut}}{\log MFI_{WT} - \log AF_{WT}} \tag{2}$$

where AF (autofluorescence) is the MFI of the corresponding cell line in the same experiment in the absence of the fluorescent chemokine.

### Calcium flux assays
MOLT-4, HEK-CCR9 and respective mutant cells were seeded at $3 \times 10^4$ cells/well in 384-well black clear flat bottom plates. 4 h later, cells were loaded with Fluo-8 calcium-sensitive fluorescent dye according to the manufacturer's instructions.

Fluorescence signals (excitation, 490 nm; emission, 525 nm) were recorded using an FDSS instrument (HAMAMATSU). In the agonist mode, signals were recorded before and after the addition of WT CCL25 or CCL25 analogs diluted to defined concentrations in assay buffer (1% BSA and 25 mM HEPES). For assessing the antagonist activity of phage display-derived CCL25 variants (or their capacity to desensitize the receptor), measurements were made following the addition of 100 nM CCL25 5 minutes later.

For each well, the recorded fluorescence signals were window-averaged using a rolling window over 10 acquisition points (5 sec), then divided by the baseline signal of the same well acquired just before the corresponding treatment, and subsequently divided by the fluorescence values recorded for cells treated with vehicle only during the first (agonist mode) or both (antagonist mode) injections. Agonist responses were expressed as:

$$Ca^{2+}\ signal = \max\left(\left[\frac{Fluorescence(t)}{Mean\ Fluorescence_{t_{inje}}}\right]_{agonist} \bigg/ \left[\frac{Fluorescence(t)}{Mean\ Fluorescence_{t_{inje}}}\right]_{Buffer}\right) \tag{3}$$

where "$t_{inj}$" is the time of injection of agonist or buffer.

## BRET assay for arrestin recruitment

HEK-CCR9-RLuc8 YFP-Arr3 and respective mutant cells were detached with 0.5 mM EDTA, seeded in 384-well black, clear flat-bottom plates at $2 \times 10^4$ cells per well in 30 µL per well of FluoroBrite™ DMEM, 10% FBS, and incubated overnight at 37 °C, 5% $CO_2$. Cells were then incubated for 10 minutes in BRET buffer (0.14 M NaCl, 6 mM KCl, 2 mM $MgSO_4$, 15 mM HEPES, 1 g/L glucose, 1% BSA) containing 20 µM coelenterazine h and then stimulated with either chemokines at defined concentrations diluted in BRET buffer or BRET buffer alone. Luminescence at 540 nm and 470 nm was measured over 10 min using an FDSS instrument (HAMAMATSU) after agonist or buffer injection, with BRET ratio defined as:

$$BRET\ ratio(t) = \frac{em_{540}(t)}{em_{470}(t)} \tag{4}$$

For each well, the recorded BRET ratios were window-averaged using a rolling window over 10 acquisition points (5 sec). Agonist responses were defined as area under the curve (AUC) of BRET signal:

$$BRET\ signal = \left( \left[ \frac{BRET\ ratio(t)}{BRET\ ratio(t_{injection})} \right]_{agonist} \bigg/ \left[ \frac{BRET\ ratio(t)}{BRET\ ratio(t_{injection})} \right]_{Buffer} \right) \tag{5}$$

Basal association (in the absence of agonist stimulation) was measured between the RLuc8-tagged receptor and YFP-tagged Arr3 by BRET (Supplementary Fig. 17A). YFP-Arr3 acceptor expression levels were assessed by flow cytometry and correlated with donor (WT or mutant CCR9-RLuc8) luminescence and basal BRET (Supplementary Fig. 17B).

## BRET Gαi/Gβγ dissociation assays

HEK-CCR9-WT, HEK-CCR9-T208A and HEK-CCR9-Q267A cells were co-transfected with Gαi(91)-Rluc2, mVenus-Gβ1 and untagged Gγ2 (in a 1:5:5 ratio) in a 6 well-plate with jetPRIME® reagent, according to the manufacturer's protocol. After 24 h, cells were detached with 0.5 mM EDTA and seeded at $2 \times 10^4$ cells/well in 384-well / flat bottom plates in 30 µL/well of FluoroBrite™ DMEM. Cells were then incubated for 10 minutes in 1X PBS containing 20 µM coelenterazine h followed by stimulation with either CCL25 at defined concentrations in BRET buffer or BRET buffer alone for 10 min. Signals were measured using an FDSS instrument (HAMAMATSU). Agonist responses were defined as the area over the curve (AOC) of BRET signal (Eq. 4). Expression levels of mVenus-Gβ1 were quantified by flow cytometry.

## Data analysis

Data normalization, collation, and statistical analyses were performed in GraphPad Prism 10.0. For chemokine binding to CCR9 mutants, binding ratios (Eq.1) were log-transformed and evaluated using one-way ANOVA with post-hoc tests and Holm-Šídák's correction for multiple comparisons. For CCR9 mutant signaling assays, areas under concentration response curves (AUCRC) were calculated using log-transformed concentrations, divided by WT AUCRC from the same experiment, and similarly evaluated using repeated measure one-way ANOVA with post-hoc tests and Holm-Šídák's correction for multiple comparisons with a single pooled variance. The p-values for all measured outputs are provided in Supplementary Tables 1, 2 and 4. All statistical tests in this study were two-sided, and p-values ≤ 0.05 were considered statistically significant. Radar plots were constructed using python and matplotlib[112].

## Reporting summary

Further information on research design is available in the Nature Portfolio Reporting Summary linked to this article.

## Data availability

All data needed to evaluate the conclusions in the study are present in the manuscript or in its Supplementary Materials, Supplementary Data. Source data are provided with this paper.

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

## Acknowledgements

This work was supported by NIH grants R21 AI149369, R21 AI156662 (to I.K.) and R01 AI161880, R01 GM136202 (to I.K. and T.M.H.) by Swiss National Science Foundation grant 310030_184828 (to O.H.) and by Innosuisse project grant 46008.1 IP-LS (to O.H.).

## Author contributions

O.H. and I.K. designed the study. I.D.M.P. generated and assessed the mutant receptor cell lines. J.R.D.D. built and analyzed structural models of receptor–chemokine complexes. I.D.M.P., N.C., and I.K. performed statistical analyses. G.T. created radar plot visualizations. M.P.-B. and K.B.A. synthesized wild-type and mutant chemokines. K.B.A. and N.C. performed CCL25 molecular evolution studies. O.H., I.K., and T.M.H. acquired funding. I.D.M.P., J.R.D.D., I.K., and O.H. wrote the manuscript. I.D.M.P., J.R.D.D., T.M.H., I.K., and O.H. revised and edited the manuscript. All authors reviewed and approved the final version of the manuscript.

## Competing interests

I.P., N.C, M.P., K.A,. and O.H. are inventors of the CCL25 analogs described in the manuscript. The CCL25 analogs were discovered as part of a collaboration agreement between the University of Geneva and Orion Biotechnology; O.H. is a co-founder and employee of Orion Biotechnology. Technology licensed by the University of Geneva to Orion Biotechnology was used in the discovery of CCL25 analogs described in this study. T.M.H. is a cofounder of Lassogen Inc. and serves on the Scientific Advisory Boards of Abilita Bio, Abalone Bio and Aikium Inc. The terms of these arrangements have been reviewed and approved by the University of California, San Diego in accordance with its conflict-of-interest policies. The remaining authors declare no competing interests.
