## [Transparent Peer Review file · Nature Communications]

Noncanonical roles of chemokine regions in CCR9 activation revealed by structural modeling and mutational mapping

Corresponding Author: Dr Oliver Hartley

Version 0:

Reviewer comments:

Reviewer #2

(Remarks to the Author)

In the paper, 'Noncanonical roles of chemokine regions in CCR9 activation revealed by structural modeling and mutational mapping' by De Magalhaes Pinheiro et al. the authors create computational models validated through binding and signaling assays to understand and manipulate the CCR9-CCL25 interaction. The authors find that, unlike previously characterized chemokine-chemokine receptor complexes, CCL25 utilized its 30s loop to facilitate the binding and activation of CCR9. These data challenge the two-step, two-site model of chemokine receptor activation by demonstrating that the CCL25 N-terminus plays a relatively minor role in receptor activation.

The authors used phage display to optimize the CCL25 N-terminal sequence for superior binding and signaling potency. The evolved superagonist [1P6]CCL25 is an notable demonstration of the capacity for protein engineering of chemokine variants with potential utility.

The conclusions of the paper attempt to integrate their CCR9 model into the broader context of solved chemokine receptor complexes. The authors draw parallels between their structural model and the ACKR3-CXCL12 interaction, but fail to acknowledge the more relevant similarities to the cryoEM structure of a CCL20-CCR6 complex as detailed below. Thus, multiple exceptions to the "N-terminal message" paradigm have already been characterized in detail, and their model for CCL25 activation simply demonstrates that the old paradigm applies only to a subset of chemokine-receptor pairs and illustrates the extent to which the molecular pharmacology of the chemokine network remains to be mapped.

Overall, the data throughout the manuscript are well-presented, visually compelling, and generally support the authors' conclusions. However, closer inspection reveals certain areas where the data presentation and methodological details are lacking. This may lead readers to question some of the conclusions of the paper. Addressing these aspects would strengthen the manuscript's impact and the reliability of the conclusions.

With appropriate revisions this manuscript would be a valuable contribution to the field of chemokine receptor biology and would be a strong candidate for publication in Nature Communications. A detailed list of comments is provided below:

Major Concerns:

1. Certain aspects of the data make it difficult to trust the conclusion of the manuscript:

a. Binding Assay

i. The binding assay used throughout the manuscript only consists of 4 data points. Furthermore, based on the description in the methods, only the max concentration point is used in the calculation of affinity. In none of the data is a maximum or minimum response established, so the data is rather qualitative rather than quantitative. This reduces confidence in the reported values. This could be addressed with a better description of how this unusual data is accurately quantifying binding.

ii. The sensitivity of the binding assay is rather low. Comparison of a parental (no CCR9) to a transfected system is only a change of 0.5 log (or a 5-fold shift). Authors discuss a lack of binding, but these ligands are just shifting beyond 5-fold. This weakness should be discussed.

b. Chemokine concentrations in manuscript

i. Throughout the manuscript, the concentrations of chemokines that are used are quite low. These concentrations include

the functional range of CCL25, but are not helpful for quantifying the results of CCR9 and CCL25 mutants.

ii. Assays throughout the paper (except figure 6) are not fit to a curve, which would improve the quantification of the data. Curve fitting would be rather difficult given the low chemokine concentration, is this why it is not used?

iii. Data in Figure 6-A,B,C/S16 are fit to curves, have 8 datapoints, a maximum, a minimum, and are a very reliable dataset.

c. BRET Assay

i. Why are data throughout fit to area under the curve of a BRET experiment? This analysis is an incomplete picture of the signaling response as it removes the impact of potency AND efficacy changes. Please explain this choice or plot the BRET assays in a traditional Δ BRET Ratio format.

2. Insufficient details in the methodology

a. Modeling

i. Please include the protein sequences that were used to generate models.

ii. In several instances, AlphaFold provides multiple models without any heterogeneity. Did all models look the same after initial modeling?

iii. Why was only the Calpha position minimized? Does this minimization alone fix all clashes that arise from the pQ substitution?

iv. Suggestion: Authors could include a schematic of the modeling pipeline and how they arrived at a final model. This would make it easier for the reader to understand the process.

b. How were the chemokines refolded? Buffers?

c. Binding assay, as noted above – include more detail as to how this serves as a quantitative method.

3. Evaluation of CCL25 and CCR9 mutagenesis results

a. The relevance of D6 is perhaps inappropriately minimized, since it is conserved in several other CC chemokines (15, 19, 20, 21, 23 – shown in Fig S5A but not mentioned in the text) and its importance for CCL20 binding to CCR6 is well documented (PMID 30114340 and ref 45).

b. The novelty of the 'non-canonical' nature of the CCL25-CCR9 interface is overstated, particularly given that the assertion (Discussion para 2) "The only other known exception to the "N-terminal message" paradigm involves the atypical chemokine receptor ACKR3 which retains its functional response (Arr3 recruitment) to CXCL12 N-terminal mutants" is simply incorrect. As Wasilko (ref 45) notes "The N-terminal length of various chemokines has been shown to be critical for receptor activation. However, CCL20 chemotactic potency has demonstrated good tolerance toward N-terminal truncations." Wasilko cites the earlier (2018) work of Riutta et al, who noted that "We show that CCL20 tolerates manipulation of its N-terminus with minimum impairment to ligand potency as long as Asp5 is present. Our findings diverge from the two-step/two-site paradigm"

Consequently, the manuscript should be revised to correct claims of novelty/uniqueness and cite the relevant earlier work on CCL20, where a non-canonical reliance on chemokine N-terminal composition was first demonstrated for a chemotactic (i.e. non-atypical) receptor. Likewise, Szpakowska [PMID 29272550] performed the first systematic analysis of the effect of N-terminal chemokine truncations demonstrating ACKR3's permissiveness relative to the 'typical' receptors CXCR3 and CXCR4 with which it shares the ligands CXCL11 and CXCL12.

c. The data presented in the manuscript sufficiently detail the importance of CCL25 residues and CCR9 residues in binding, but do not provide experimental evidence linking key interface residues of the chemokine and receptor. If there was a way to validate the interface of these two, it would strengthen the paper. This could occur through charge swap mutations, cysteine crosslinking, etc. Of course, these experiments are difficult and this is not a requirement.

Minor Concerns:

1. Aspects of the manuscript could be visually improved to make it easier to understand.

a. Residues should be labeled throughout with BW nomenclature (and vice versa with CCR9 numbering)

i. BW used well in the text, but missing from every figure.

ii. Page 10 line 4. Used BW (6.48 and 6.51) but missing CCR9 residues.

iii. Include residue labels in 2BCD/8DEF.

b. Radar plots are inconsistent

i. In Figure 2A/8ABC mutants are on the outside and assays are on the inside. Figure 3DG/4D/5D the plots are flipped. Keep mutants on the outside and color by assay.

c. Bar graphs

i. Figure 6D has the direction of a negative shift flipped from all other bar graphs. This was confusing.

ii. Bar graphs would be easier to read if WT was 0 and shifts were + or – log values.

d. Errors in plots

i. Why is Fig S5 listed in the manuscript before Fig S1?

ii. Fig S8f – Is this a BRET n=3 or Calcium Flux n=3?

2. Add redundant details in figures/tables/legends

a. Include Z= pQ in every table

3. Clarification

a. "20 chemokine structures vs 40 chemokine structures" – is this comparing Cryo-EM to Xray structures?

b. T stacks – niche term, please explain in more detail

Reviewer #3

(Remarks to the Author)

Reviewer #4

(Remarks to the Author)

Pinheiro et al provided a detailed exploration of the structural and functional interactions between the CCR9 receptor and its endogenous agonist, CCL25. It highlights the unique role of the 30s loop of CCL25 in receptor activation and signaling, challenging the conventional view that the N-terminus of chemokines is primarily responsible for receptor engagement. The authors present compelling evidence suggesting that the 30s loop plays a central role in receptor activation, which could have significant implications for drug design and therapeutic strategies targeting CCR9.

The data presented in the manuscript appear robust and technically sound. The authors utilize a well-rounded approach that includes structural modeling, mutagenesis, and pharmacological assays to support their conclusions. The experiments are well-controlled, although further elaborations would enhance the validity of their claims.

The methodologies employed are appropriate for the research questions posed. The use of AlphaFold 2 (AF2) for structural modeling is commendable, yielding models that are consistent with the existing knowledge of GPCR-peptide complexes. However, providing additional details on the criteria used to evaluate the generated models and the potential limitations of AF2 in capturing dynamic states would strengthen the presentation. It is crucial to acknowledge the limitations of AF2, particularly regarding its ability to capture dynamic states and conformational flexibility inherent to GPCR signaling. The nature of GPCR activation involves significant conformational changes that are often not fully represented in static structural models. The authors do not adequately address the role of dynamics in the signaling process, which is a critical aspect of GPCR function. For instance, integrating molecular dynamics simulations could provide valuable insights into the conformational transitions that occur during receptor activation and how these transitions influence signaling outcomes. By incorporating a discussion on the dynamic nature of the CCR9-CCL25 interaction, the authors could enrich the understanding of the signaling mechanisms at play, thereby strengthening the overall impact of their findings. Highlighting how dynamic conformational states relate to functional outcomes would significantly enhance the context and relevance of the study in the broader field of GPCR research. It is disappointing that the authors did not discuss the impact of dynamics in the introduction, as this omission limits the contextual foundation of the study. Upon reviewing the manuscript as a whole, it appears that the authors may have chosen to minimize the emphasis on dynamics, which could be seen as an attempt to mitigate the limitations of their computational approach in capturing dynamic processes.

1- The authors should elaborate on the limitations of AlphaFold 2 in terms of its ability to capture the full spectrum of GPCR dynamics. A critical assessment of AF2's predictive capabilities, particularly concerning conformational flexibility and the entropic contributions to binding, would provide a more balanced view of the modeling approach. This could include a discussion on how the integration of experimental data, such as molecular dynamics or other biophysical techniques, could complement AF2 predictions.

2- The authors should elaborate on their choice of specific experimental techniques used in conjunction with the modeling approach. A detailed explanation of how these techniques were selected to complement the structural models and provide validation of their findings would enhance the manuscript. This discussion should include how each technique contributes to understanding the CCR9-CCL25 interaction and signaling mechanisms.

3- The authors do discuss structural comparisons between their findings and known GPCR-chemokine complexes, such as CCR1, CCR2, and CCR5. However, while they mention these structural similarities, the discussion lacks depth in contextualizing their results within the broader landscape of GPCR signaling. Specifically, they do not adequately compare the CCR9-CCL25 interaction with other GPCR-chemokine complexes in terms of functional outcomes, signaling biases, or unique aspects of their system. To enhance the manuscript, the authors should expand on this comparison, highlighting not only the structural similarities but also the functional implications of their findings. Integrating a discussion on how their results relate to existing literature on GPCR signaling biases would significantly enrich the manuscript.

4- The authors should consistently adhere to the Ballesteros-Weinstein (BW) numbering system throughout the manuscript. This consistency is crucial for clarity and ease of reference, especially when discussing specific amino acid positions in the context of receptor mutations and interactions.

5- The authors touched upon signaling biases related to the identified mutations, but they could further explore how these mutations might influence G protein versus arrestin coupling preferences.

6- The authors should consider tuning down their claims throughout the manuscript. Many of their observations are primarily based on AlphaFold 2 models and point mutations, which may not capture the full complexity and specificity of the CCR9-CCL25 interaction. It is important to acknowledge the limitations of these methodologies in drawing broad conclusions.

7- The authors generated alanine substitution mutants to probe functional significance, which is a common approach; however, relying solely on alanine mutations may oversimplify the interpretation of functional roles. They could discuss the potential limitations of this approach and consider a broader range of substitutions (e.g., larger residues or charged residues) to provide more insight into the specific roles of the targeted residues. Additionally, it could benefit from a more detailed analysis of the specific residues selected for mutation. For example, how do these residues interact with CCL25, and what are their specific roles in the signaling process? The rationale for selecting specific residues for mutation should be better articulated. For example, while the authors group mutations by spatial location and impact, it would be beneficial to explain why certain residues were prioritized for this analysis and how they were hypothesized to influence chemokine binding and signaling

8- The text could better clarify the functional implications of the mutations at S2075.31 and T2085.32. While the effects of these mutations on binding and signaling are described, a more explicit connection between the observed changes and the underlying molecular mechanisms would enhance understanding.

9- In Figure 1, part D utilizes BW labeling for certain residues, while parts E and F do not employ the same labeling approach. Consistent visualization and labeling across all parts of a figure are crucial for clarity and ease of understanding. Furthermore, I can see this inconsistency in labeling of all figures in the manuscript.

10- In parts B, C, and D of Figure 2, residues are presented as balls without any labels. This presentation limits the figure's informational value, as readers cannot relate the individual balls to specific residues. In Figure 8, there is a similar problem. To enhance the clarity and utility of the figure, it is essential to label each ball appropriately. A redesign of this figure to include clear labeling or an alternative presentation that maintains readability would significantly improve its effectiveness.

11- Parts H, I, and J of Figure 3 are excessively crowded, which significantly hinders the viewer's ability to interpret the interactions effectively. The representation of interactions as dots contributes to the confusion, obscuring the underlying purpose of the figure. To improve clarity and comprehension, I recommend redesigning these section.

Reviewer #5

(Remarks to the Author)

The manuscript describes hybrid computational experimental approach to elucidating mechanism of CCR9-CCL25 interaction and CCR9 activation and provides insight in structure based drug discovery. Overall approach is sound and it is nice that the results of computational modeling is followed by systematic mutational studies to support the model. The manuscript is well written and easy to follow and provides a lot useful information on the important system. Below are few comments which might be useful to include to the final version of the manuscript.

1) It would be good to store/report somewhere final suggested model in pdb and potentially pse (pymol session) format with mutated residues highlighted and color coded for the agreement with experimental data. May be I missed it somewhere but I can't see it anywhere.

2) Alternative models are also potentially interesting to report together with data to see the level of uncertainty.

3) I see the authors used their own confidence model for the choice of the model - what was original AF2 Multimer confidence at the interface of the best and selected model/

4) The authors run 5 seeds/5models (ensemble of 25 structures) for AF2 - while it could be enough for easy cases - I would run 100 seeds 5 models (as a matter of fact this is what community finds the best as well as DeepMind itself) - would be good to confirm that most confident structure doesn't change.

5) Since AF3 server is available I would also suggest to run multiple seeds of AF3 - may be 20 (as currently available from single account)

I believe extended analysis of the models /reporting interface confidence and providing analysis of models in the supplement together with reporting final model together with overlapped experimental results would greatly improve the manuscript

Version 1:

Reviewer comments:

Reviewer #2

(Remarks to the Author)

We thank the authors for their thorough and thoughtful responses to the comments from all reviewers.

The revised manuscript, 'Noncanonical roles of chemokine regions in CCR9 activation revealed by structural modeling and mutational mapping,' stands out as one of the few studies to rigorously validate a computational model with experimental approaches. This validation and the novel insights into the CCL25 binding mode make it a strong contribution to the

chemokine field.

Following careful review, we find that the authors have effectively addressed the limitations of the original manuscript through detailed and constructive responses. We recommend this manuscript for publication.

Our only remaining recommendation is that the authors include the raw calcium flux and BRET assay data to enhance reproducibility and transparency. We also suggest that the authors include the reviewer responses as supplementary material, as they provide important insights into the methodology and its inherent limitations.

Reviewer #3

(Remarks to the Author)

Reviewer #4

(Remarks to the Author)

Thank you for the revisions. The manuscript is now more clearly presented in terms of methodology, and the interpretation of the results has improved with a more balanced discussion, particularly in relation to the limitations of AlphaFold2. The inclusion of a brief acknowledgment regarding the restricted ability of AlphaFold2 to model dynamic aspects of GPCR activation helps contextualize the findings, though this remains an inherent limitation of the computational approach.

The mutagenesis and structural modeling are well-integrated, and the logic behind residue selection is now better explained. However, the absence of direct experimental validation of predicted contacts still leaves room for ambiguity. While you address this by citing prior experience and noting the technical challenges, it would be useful to be more explicit in the manuscript about how future work might aim to close this gap. For example, stating whether charge-reversal or crosslinking strategies are being considered or how feasible they might be in this system could help orient readers and frame the current findings as part of an ongoing effort rather than a complete resolution.

In its current form, the manuscript is more solid, but this clarification would improve the framing of the conclusions and more transparently communicate the limitations and opportunities ahead.

Reviewer #5

(Remarks to the Author)

The authors did a great job on the revision. All requested additional runs with AF2 and AF3 are done, and deposited. I think the manuscript is good now from my viewpoint

Responses to critiques from Reviewer #2

In the paper, 'Noncanonical roles of chemokine regions in CCR9 activation revealed by structural modeling and mutational mapping' by De Magalhaes Pinheiro et al. the authors create computational models validated through binding and signaling assays to understand and manipulate the CCR9-CCL25 interaction. The authors find that, unlike previously characterized chemokine-chemokine receptor complexes, CCL25 utilized its 30s loop to facilitate the binding and activation of CCR9. These data challenge the two-step, two-site model of chemokine receptor activation by demonstrating that the CCL25 N-terminus plays a relatively minor role in receptor activation.

The authors used phage display to optimize the CCL25 N-terminal sequence for superior binding and signaling potency. The evolved superagonist [1P6]CCL25 is a notable demonstration of the capacity for protein engineering of chemokine variants with potential utility.

The conclusions of the paper attempt to integrate their CCR9 model into the broader context of solved chemokine receptor complexes. The authors draw parallels between their structural model and the ACKR3-CXCL12 interaction, but fail to acknowledge the more relevant similarities to the cryoEM structure of a CCL20-CCR6 complex as detailed below. Thus, multiple exceptions to the "N-terminal message" paradigm have already been characterized in detail, and their model for CCL25 activation simply demonstrates that the old paradigm applies only to a subset of chemokine-receptor pairs and illustrates the extent to which the molecular pharmacology of the chemokine network remains to be mapped.

Overall, the data throughout the manuscript are well-presented, visually compelling, and generally support the authors' conclusions. However, closer inspection reveals certain areas where the data presentation and methodological details are lacking. This may lead readers to question some of the conclusions of the paper. Addressing these aspects would strengthen the manuscript's impact and the reliability of the conclusions.

With appropriate revisions this manuscript would be a valuable contribution to the field of chemokine receptor biology and would be a strong candidate for publication in Nature Communications. A detailed list of comments is provided below:

Response: We appreciate the thoughtful and positive overall assessment of our work and acknowledge the indicated omissions. The modifications that we made to address these points are detailed below.

Major Concerns:

1. Certain aspects of the data make it difficult to trust the conclusion of the manuscript:
 - a. Binding Assay
 - i. The binding assay used throughout the manuscript only consists of 4 data points. Furthermore, based on the description in the methods, only the max concentration point is used in the calculation of affinity. In none of the data is a maximum or minimum response established, so the

data is qualitative rather than quantitative. This reduces confidence in the reported values. This could be addressed with a better description of how this unusual data is accurately quantifying binding.

ii. The sensitivity of the binding assay is rather low. Comparison of a parental (no CCR9) to a transfected system is only a change of 0.5 log (or a 5-fold shift). Authors discuss a lack of binding, but these ligands are just shifting beyond 5-fold. This weakness should be discussed.

b. Chemokine concentrations in manuscript

i. Throughout the manuscript, the concentrations of chemokines that are used are quite low. These concentrations include the functional range of CCL25, but are not helpful for quantifying the results of CCR9 and CCL25 mutants.

ii. Assays throughout the paper (except figure 6) are not fit to a curve, which would improve the quantification of the data. Curve fitting would be rather difficult given the low chemokine concentration, is this why it is not used?

iii. Data in Figure6-A,B,C/S16 are fit to curves, have 8 datapoints, a maximum, a minimum, and are a very reliable dataset.

Response: The primary aim of our binding assay was to assess the relative impact of CCR9 mutations on CCL25 binding, rather than to determine absolute affinities. For this comparative purpose, a simplified assay using four concentrations of fluorescently labeled CCL25 was both appropriate and informative.

We note that chemokines, including CCL25, bind not only to their receptors but also to abundant cell surface glycosaminoglycans (GAGs) (Crijns et al. 2020 doi:10.3389/fimmu.2020.00483). This receptor-independent binding becomes increasingly prominent at higher nanomolar concentrations, which explains why:

- CCR9-negative cells still show detectable binding signals;
- the increase in signal on CCR9-expressing cells is modest (~0.5 log);
- binding curves often fail to reach a clear upper plateau.

We avoided higher ligand concentrations to prevent dominance of the GAG-mediated signal. While using lower concentrations could enhance receptor specificity, accurate measurements at those levels typically require radioiodinated chemokines. Fluorescent ligands, though less sensitive, remain suitable for evaluating relative differences across receptor mutants.

Accordingly, we used 300 nM as the top concentration, as it consistently provided the best signal-to-background ratio. Although this precluded curve fitting for the CCR9 mutant panel, it enabled practical comparisons across mutants. The full concentration range data are provided in the supplementary material.

By contrast, experiments involving CCL25 mutants on WT CCR9 were feasible to perform at scale, and thus allowed for full concentration-response curves (Fig. 6 and S16, now Fig. S20). In the revised manuscript, we have clarified this methodological distinction and explicitly acknowledged the limitations of the binding assay in providing quantitative affinity estimates.

c. BRET Assay

i. Why are data throughout fit to the area under the curve of a BRET experiment? This analysis is an incomplete picture of the signaling response as it removes the impact of potency AND efficacy changes. Please explain this choice or plot the BRET assays in a traditional Δ BRET Ratio format.

Response: We used a two-step data reduction process to enable comparative analysis across multiple CCR9 mutants and assay readouts. First, time-dependent Δ BRET traces obtained for each chemokine concentration were integrated to yield area under the curve (AUC) values. These were plotted as concentration–response curves in the main and supplementary figures (e.g., Figs. 3C, 4C, 5C, 7D, S10–12, S15, S22–23, S27–29).

Second, to create the radar plots, we further condensed these concentration–response curves into a single metric per mutant per assay, using the area under the concentration–response curve (AUCRC). This allowed us to represent multi-dimensional signaling data in a compact and interpretable visual format.

We fully agree with the reviewer that AUCRC values merge effects on potency and efficacy, and therefore cannot distinguish between them. For this reason, we provide the full concentration–response profiles in the supplementary figures, allowing the reader to examine potency and efficacy effects separately. In the revised manuscript, we have clarified this methodology and explicitly directed readers to the full data.

2. Insufficient details in the methodology

a. Modeling

i. Please include the protein sequences that were used to generate models.

ii. In several instances, AlphaFold provides multiple models without any heterogeneity. Did all models look the same after initial modeling?

iii. Why was only the C α position minimized? Does this minimization alone fix all clashes that arise from the pQ substitution?

iv. Suggestion: Authors could include a schematic of the modeling pipeline and how they arrived at a final model. This would make it easier for the reader to understand the process.

Response:

(i) the AA sequences used for modeling are included in the corresponding Methods section

(ii) for CCR9-CCL25, AlphaFold2 Multimer reproducibly generates an ensemble with conformational heterogeneity; moreover, the generated models span a range of receptor activation states (**Fig. 4**). We attribute this phenomenon to the fact that for this receptor, a fully inactive structure (PDB 5lwe) was available in the AF2 training set (2021 PDB), as well as active and inactive structures of chemokine complexes with homologous receptors. Because AF2 model heterogeneity is often correlated with protein dynamics (e.g. Guo *et al.* 2022 doi: 10.1038/s41598-022-14382-9), we made use of this heterogeneous conformational ensemble for understanding the effects of selected mutations.

(iii) all atoms of the chemokine and side-chain atoms of the receptor were minimized. The positional restraints were only applied to C α atoms of the chemokine to allow the remaining atoms to move freely and form the most energetically favorable conformations and contacts. The nature and strength of the restraints are such that they always allow some movement even for the restrained atoms and, consequently, the resolution of minor clashes. There are no clashes in the final models of any of the presented complexes.

(iv) a supplemental figure with the modeling workflow is included in the revised manuscript.

b. How were the chemokines refolded? Buffers?

Response: Chemokines were refolded in a glutathione redox buffer. We have added more detail in the methods section of the revised manuscript

c. Binding assay, as noted above – include more detail as to how this serves as a quantitative method.

Response: Please see response to 1a above.

3. Evaluation of CCL25 and CCR9 mutagenesis results

a. The relevance of D6 is perhaps inappropriately minimized, since it is conserved in several other CC chemokines (15, 19, 20, 21, 23 – shown in Fig S5A but not mentioned in the text) and its importance for CCL20 binding to CCR6 is well documented (PMID 30114340 and ref 45).

Response: We agree that D6 is a relatively conserved residue among several CC chemokines (e.g., CCL14, CCL15, CCL17, CCL19, CCL20, CCL21, CCL23, and CCL25), and that its functional role has been demonstrated in other contexts. In particular, Riutta et al. (doi:10.1002/JLB.1VMA0218-049R; now cited in the revised manuscript) showed that the D5A mutation in CCL20 significantly impaired CCR6-mediated calcium mobilization (~2 log reduction) and moderately weakened binding (~0.5 log), indicating that D5 is an important activation determinant in that system.

In contrast, in our CCR9–CCL25 system, the analogous D6A mutation caused only a minor reduction in chemokine potency, while mutations in the 30s loop had a much stronger effect (Fig. 6). Although we cannot directly compare this with CCL20, since the earlier study did not investigate 30s loop mutations, these findings suggest that the functional role of this conserved residue varies depending on the receptor–chemokine context.

We have included this discussion in the revised manuscript to clarify the specific, context-dependent contribution of D6 to CCR9 activation.

- b. The novelty of the ‘non-canonical’ nature of the CCL25-CCR9 interface is overstated, particularly given that the assertion (Discussion para 2) “The only other known exception to the “N-terminal message” paradigm involves the atypical chemokine receptor ACKR3 which retains its functional response (Arr3 recruitment) to CXCL12 N-terminal mutants” is simply incorrect. As Wasilko (ref 45) notes “The N-terminal length of various chemokines has been shown to be critical for receptor activation. However, CCL20 chemotactic potency has demonstrated good tolerance toward N-terminal truncations.” Wasilko cites the earlier (2018) work of Riutta et al, who noted that “We show that CCL20 tolerates manipulation of its N-terminus with minimum impairment to ligand potency as long as Asp5 is present. Our findings diverge from the two-step/two-site paradigm”. Consequently, the manuscript should be revised to correct claims of novelty/uniqueness and cite the relevant earlier work on CCL20, where a non-canonical reliance on chemokine N-terminal composition was first demonstrated for a chemotactic (i.e. non-atypical) receptor. Likewise, Szpakowska [PMID 29272550] performed the first systematic analysis of the effect of N-terminal chemokine truncations demonstrating ACKR3’s permissiveness relative to the ‘typical’ receptors CXCR3 and CXCR4 with which it shares the ligands CXCL11 and CXCL12.

Response: We have acknowledged this point in the Discussion section of the revised manuscript, additionally citing the work by Szpakowska *et al.*:

- “Previous studies have identified some deviations from the two-site model, such as the tolerance of CCR6-CCL20 or the atypical chemokine receptor, ACKR3, to modifications at the distal N-terminus of the chemokine. However, these studies still highlighted residues at the proximal N-terminus as critical contributors to receptor activation”.

- c. The data presented in the manuscript sufficiently detail the importance of CCL25 residues and CCR9 residues in binding, but do not provide experimental evidence linking key interface residues of the chemokine and receptor. If there was a way to validate the interface of these two, it would strengthen the paper. This could occur through charge swap mutations, cysteine crosslinking, etc. Of course, these experiments are difficult and this is not a requirement.

Response: We agree that charge-swap mutations and disulfide crosslinking are powerful approaches for validating intermolecular contacts, and we have used them successfully in

previous studies (Stephens et al., 2020; doi:10.1126/scisignal.aay5024; Ngo et al., 2020; doi:10.1371/journal.pbio.3000656). However, these experiments are technically demanding — particularly in the case of CCR9–CCL25, where the number and spatial distribution of candidate contact points would make systematic application of these methods challenging. Given the availability of high-resolution cryo-EM structures for related chemokine receptor complexes (e.g., CCR1, CCR2, CCR5, CCR6), and the ability of AlphaFold2 to generate activation-relevant conformational ensembles, we felt that additional crosslinking experiments would provide only incremental value in this context. We appreciate the reviewer’s suggestion and their recognition that these experiments, while informative, are not essential for the present study.

Minor Concerns:

1. Aspects of the manuscript could be visually improved to make it easier to understand.
 - a. Residues should be labeled throughout with BW nomenclature (and vice versa with CCR9 numbering)
 - i. BW used well in the text, but missing from every figure.
 - ii. Page 10 line 4. Used BW (6.48 and 6.51) but missing CCR9 residues.
 - iii. Include residue labels in 2BCD/8DEF.

Response:

- (i) In the revision, BW indices have been added to all figures.
- (ii) This has been clarified in the revision. In sentences that discuss roles of homologous residues not only CCR9 but also in other GPCRs (including the sentence mentioned by the Reviewer), we prefer using the BW index of that residue and not its specific value for CCR9.
- (iii) Residue labels and BW indices have been added to Figs. 2 and 8 in the revision.

- b. Radar plots are inconsistent
 - i. In Figure 2A/8ABC mutants are on the outside and assays are on the inside. Figure 3DG/4D/5D the plots are flipped. Keep mutants on the outside and color by assay.

Response: This was intentional. Figures 2A and 8ABC are designed to emphasize the asymmetric nature of the signaling profiles across different assays (2A), and the differences between native CCL25 and the superagonist analog (8ABC). For this reason, each contour in these figures represents a different assay. By contrast, Figures 3DG, 4D, and 5D are intended to highlight differences between receptor mutants, with each contour corresponding to a different mutant — a format commonly used to represent signaling bias (e.g., Narayanan et al. 2020; doi:10.1016/j.bmc.2019.115237). Flipping the axes in Figures 3DG, 4D, and 5D as suggested would undermine this distinction and render those figures effectively redundant with Figure 2A. We hope that based on this clarification the Reviewer agrees with our rationale.

- c. Bar graphs
 - i. Figure 6D has the direction of a negative shift flipped from all other bar graphs. This was confusing.
 - ii. Bar graphs would be easier to read if WT was 0 and shifts were + or – log values.

Response:

- i. As recommended by the reviewer we have reverted this graph so that negative shifts in signaling activity result in downward bars.
- ii. We have opted to maintain representation as a ratio expressed on a log scale, in order to retain consistency with the other bar graphs, which are presented in this way.

- d. Errors in plots
- i. Why is Fig S5 listed in the manuscript before Fig S1?
- ii. Fig S8f – Is this a BRET n=3 or Calcium Flux n=3?

Response:

- (i) The order in which supplemental figures were called out in the original manuscript was correct: figures S1-S4 were cited in the Methods which in the original version preceded the Results. In the Revision, we changed the section order so that Results precede Methods; the supplemental figures have been reordered accordingly.
- (ii) The plot label typo on Fig S8F (now Fig S10F) has been fixed in the revised manuscript.

- 2. Add redundant details in figures/tables/legends
 - a. Include Z= pQ in every table

Response: This modification has been made to the tables concerned (S4 and S5) in the revised manuscript.

- 3. Clarification
 - a. “20 chemokine structures vs 40 chemokine structures” – is this comparing Cryo-EM to X-ray structures?
 - b. T stacks – niche term, please explain in more detail

Response:

- (a) Of the 26 receptor structures with chemokines currently in the PDB (Dec 2024), 23 have been solved by Cryo-EM. Of the 21 *active-state* receptor structures with chemokines currently in the PDB, all 21 have been solved by cryo-EM. The numbers have been updated to reflect the Dec 2024 statistics and the sentence has been reworded for clarity; it now reads “...23 chemokine complex **cryo-EM** structures of 12 receptors in the PDB as of December 2024, out of the total of 63 structures of 15 receptors by all methods...”
- (b) Sentence reworded for clarity: “...while F299^{7.35} participates in a perpendicular/T-shaped stacking with CCL25 residue F4”.

Responses to critiques from Reviewer #4

Pinheiro et al provided a detailed exploration of the structural and functional interactions between the CCR9 receptor and its endogenous agonist, CCL25. It highlights the unique role of the 30s loop of CCL25 in receptor activation and signaling, challenging the conventional view that the N-terminus of chemokines is primarily responsible for receptor engagement. The authors present compelling evidence suggesting that the 30s loop plays a central role in receptor activation, which could have significant implications for drug design and therapeutic strategies targeting CCR9.

The data presented in the manuscript appear robust and technically sound. The authors utilize a well-rounded approach that includes structural modeling, mutagenesis, and pharmacological assays to support their conclusions. The experiments are well-controlled, although further elaborations would enhance the validity of their claims.

The methodologies employed are appropriate for the research questions posed. The use of AlphaFold 2 (AF2) for structural modeling is commendable, yielding models that are consistent with the existing knowledge of GPCR-peptide complexes. However, providing additional details on the criteria used to evaluate the generated models and the potential limitations of AF2 in

capturing dynamic states would strengthen the presentation. It is crucial to acknowledge the limitations of AF2, particularly regarding its ability to capture dynamic states and conformational flexibility inherent to GPCR signaling. The nature of GPCR activation involves significant conformational changes that are often not fully represented in static structural models. The authors do not adequately address the role of dynamics in the signaling process, which is a critical aspect of GPCR function. For instance, integrating molecular dynamics simulations could provide valuable insights into the conformational transitions that occur during receptor activation and how these transitions influence signaling outcomes. By incorporating a discussion on the dynamic nature of the CCR9-CCL25 interaction, the authors could enrich the understanding of the signaling mechanisms at play, thereby strengthening the overall impact of their findings. Highlighting how dynamic conformational states relate to functional outcomes would significantly enhance the context and relevance of the study in the broader field of GPCR research. It is disappointing that the authors did not discuss the impact of dynamics in the introduction, as this omission limits the contextual foundation of the study. Upon reviewing the manuscript as a whole, it appears that the authors may have chosen to minimize the emphasis on dynamics, which could be seen as an attempt to mitigate the limitations of their computational approach in capturing dynamic processes.

Response: We thank the reviewer for the positive assessment of our work and for emphasizing the importance of protein dynamics in GPCR signaling. We share the reviewer's strong appreciation of dynamics and, in our work, pay a lot of attention to the receptor's conformational variability that mediates its dynamic transitions and signaling outcomes. In the point-by-point response below, we outline the changes that were introduced in the revision to address the reviewer's critiques.

1- The authors should elaborate on the limitations of AlphaFold 2 in terms of its ability to capture the full spectrum of GPCR dynamics. A critical assessment of AF2's predictive capabilities, particularly concerning conformational flexibility and the entropic contributions to binding, would provide a more balanced view of the modeling approach. This could include a discussion on how the integration of experimental data, such as molecular dynamics or other biophysical techniques, could complement AF2 predictions.

Response: The following revisions were made in the manuscript to address this critique:

- **Introduction:** "Moreover, in certain cases it [AF2] can generate conformational ensembles, with the structural variation across the ensemble reflecting conformational dynamics as observed in molecular dynamics (MD) simulations."
- **Discussion.** "Nonetheless, the ability of AF2 to predict structural dynamics is limited. Accordingly, certain conformational states of the complex remained inaccessible to our modeling, as did the entropic component of binding. As a consequence, we were only able to provide tentative explanations for the biased activity of three receptor mutants (T208^{5.32}A, Q267^{6.48}A and N271^{6.52}A) and the affinity improvements of selected chemokine variants (G2A, V3A, and G33A). Future studies using molecular dynamics, NMR, single-molecule FRET, or time-resolved cryo-EM may provide more detailed answers to these questions".

2- The authors should elaborate on their choice of specific experimental techniques used in conjunction with the modeling approach. A detailed explanation of how these techniques were selected to complement the structural models and provide validation of their findings would enhance the manuscript. This discussion should include how each technique contributes to understanding the CCR9-CCL25 interaction and signaling mechanisms.

Response: To address this comment, we introduced the following text:

- **Results:** The 12 mutants were characterized (**Fig. S7**) to determine the relative impact of each position on CCL25 binding and signaling. Binding studies were performed at four different concentrations of CCL25 site-specifically labeled with rhodamine at its C-terminal extremity (**Fig. S8**). To determine relative impact in binding we made use of data points from the highest concentration (300 nM), which provided the strongest discrimination between the CCR9-dependent binding signal and signal from CCR9-independent binding to cell surface proteoglycans. For signaling studies, we distinguished between G protein- and arrestin-driven activities using a calcium flux assay (downstream of G protein activation) and a BRET-based arrestin3 recruitment assay. Both assays were carried out at six different CCL25 concentrations, with the area under the concentration-response curve used to quantify signaling activity and assess the relative impact.
- **Discussion:** "...Individual receptor mutants ... differentially impact aspects of receptor function such as constitutive activity, G protein and arrestin-coupling preferences, intracellular trafficking and signaling from subcellular compartments [73, 81-85]. In our study, we addressed this challenge by assessing the impact of receptor and chemokine mutations on multiple aspects of receptor pharmacology".

3- The authors do discuss structural comparisons between their findings and known GPCR-chemokine complexes, such as CCR1, CCR2, and CCR5. However, while they mention these structural similarities, the discussion lacks depth in contextualizing their results within the broader landscape of GPCR signaling. Specifically, they do not adequately compare the CCR9-CCL25 interaction with other GPCR-chemokine complexes in terms of functional outcomes, signaling biases, or unique aspects of their system. To enhance the manuscript, the authors should expand on this comparison, highlighting not only the structural similarities but also the functional implications of their findings. Integrating a discussion on how their results relate to existing literature on GPCR signaling biases would significantly enrich the manuscript.

Response: We made two significant changes to address these suggestions in the revised manuscript:

- In **Results**, we explicitly delineated shared and unique features of the CCR9 complex as compared to the GPCR superfamily and the chemokine receptor subfamily:
 - "Superposition of the highest confidence AF2 model onto the X-ray structure of inactive CCR9 bound to vercirnon revealed that CCL25-bound CCR9 features an outward movement of the intracellular ends of the transmembrane (TM) helices 5 and 6 (**Fig. 1B**), a shared activation signature across the GPCR superfamily. Prominent TM helix rearrangements are also apparent in the orthosteric binding pocket on the extracellular side of the receptor (**Fig. 1C**): compared to inactive CCR9, the CCL25-bound conformation features a large inward movement of the extracellular end of TM5 with a concurrent outward movement of extracellular loop (ECL) 3 and the adjoining ends of TM6 and TM7, providing an opening to accommodate the chemokine. These movements are consistent with activation-associated rearrangements on the extracellular side of other chemokine receptors [40], but not necessarily characteristic of other GPCR subfamilies. Finally, and uniquely among the chemokine receptors studied so far, CCR9 activation is also accompanied by a profound (~6 Å) downward (i.e. in the intracellular direction) sliding-and-bending motion of TM5, and, to a lesser extent of TM6, relative to the rest of the TM bundle (**Fig. 1D**)".
- In the **Discussion**, we added the following text:
 - "Arrestin-versus-G protein signaling bias is a therapeutically exploitable phenomenon that has been also observed between the natural ligands of several

members of the chemokine receptor family. Unfortunately, despite recent progress, structural understanding of such bias remains elusive. Available for only three GPCRs so far, NTR1, CNR1, and OPSD, complementary experimental structures with G proteins and with arrestins have not revealed a generalizable conformational signature for signaling bias. Accordingly, future studies of the CCR9-CCL25 interaction using molecular dynamics, NMR, single-molecule FRET, or time-resolved cryo-EM may provide a more detailed explanation of the signaling biases of the mutants identified in our work”.

4- The authors should consistently adhere to the Ballesteros-Weinstein (BW) numbering system throughout the manuscript. This consistency is crucial for clarity and ease of reference, especially when discussing specific amino acid positions in the context of receptor mutations and interactions.

Response: We agree. Upon close inspection, we identified only one section in the Results where the BW indices were not used (the section related to **Fig. 2**). We corrected this omission. Regarding the figures, - in the initial submission, we chose to systematically omit the BW indices from the figures to avoid unnecessary crowdedness. However, it appears that this and other Reviewers prefer to see these indices in the figures. Therefore, in the revision, we added BW indices to all panels on all figures.

5- The authors touched upon signaling biases related to the identified mutations, but they could further explore how these mutations might influence G protein versus arrestin coupling preferences.

Response: Pronounced signaling biases have been identified with three receptor mutants: T208^{5.32}A, Q267^{6.48}A, and N271^{6.52}A. For T208^{5.32}A, we offer the following explanation of how it might affect G protein-mediated Ca²⁺ response versus arrestin coupling preferences:

- “We hypothesize that Arr3 recruitment requires this **“most active” state involving the T208^{5.32}-N36 hydrogen bond**, whereas G protein association is permissive to a range of active-like conformations of CCR9, as previously described for other GPCRs. **The loss of T208^{5.32}-N36 hydrogen bonding** in the T208^{5.32}A mutant would therefore **selectively abrogate Arr3 recruitment** with minimal impact on G protein-mediated Ca²⁺ mobilization.”

For Q267^{6.48}A and N271^{6.52}A, we admit that beyond predicting constitutive activity, our **computational models were unable to explain their striking biases**, and offer the following observations:

- “The **single helical turn** that separates Q267^{6.48} and N271^{6.52} in the CCR9 structure **harbors P269^{6.50}**, the most conserved amino acid in TM6 of class A GPCRs and the core of the TM6 kink. Moreover, mutations of Q267^{6.48} and N271^{6.52} **selectively eliminate TM6 contacts with TM7 and TM5**, respectively (**Fig. S19**). Therefore, we hypothesize that the loss of these contacts alters the conformational coupling between the binding site and the intracellular effector interface in a manner that preferentially affects Arr3 or G protein.”

Discussion of these two regions is as far as we feel able to go with respect to structural interpretation of signaling bias. Despite all the recent progress, structural and dynamic understanding of GPCR signaling bias remains a very challenging topic. Only three GPCRs (NTR1, CNR1, and OPSD, none of them a chemokine receptor) have both experimental structures with G proteins and experimental structures with arrestins. The conformational differences underlying selective GPCR coupling to G proteins vs β -arrestins have been demonstrated (through structure determination) to be receptor-specific and very subtle, with the range well within the differences between the multiple G protein-selective conformations.

6- The authors should consider tuning down their claims throughout the manuscript. Many of their observations are primarily based on AlphaFold 2 models and point mutations, which may not capture the full complexity and specificity of the CCR9-CCL25 interaction. It is important to acknowledge the limitations of these methodologies in drawing broad conclusions.

Response: The following sentences in the revised manuscript acknowledge the limitations of our approaches:

- “Beyond predicting this constitutive activity, the computational models were unable to explain the striking signaling bias of the Q267^{6.48}A and N271^{6.52}A mutants”.
- “Nonetheless, the ability of AF2 to predict structural dynamics is limited. Accordingly, certain conformational states of the complex remained inaccessible to our modeling, as did the entropic component of binding. As a consequence, we were only able to provide tentative explanations for the biased activity of three receptor mutants (T208^{5.32}A, Q267^{6.48}A and N271^{6.52}A) and the affinity improvements of selected chemokine variants (G2A, V3A, and G33A)”.
- “Arrestin-versus-G protein signaling bias is a therapeutically exploitable phenomenon that has been also observed between the natural ligands of several members of the chemokine receptor family. Unfortunately, despite recent progress, structural understanding of such bias remains elusive. Available for only three GPCRs so far, NTR1, CNR1, and OPSD, complementary experimental structures with G proteins and with arrestins have not revealed a generalizable conformational signature for signaling bias. Accordingly, future studies of the CCR9-CCL25 interaction using molecular dynamics, NMR, single-molecule FRET, or time-resolved cryo-EM may provide a more detailed explanation of the signaling biases of the mutants identified in our work”.

7- The authors generated alanine substitution mutants to probe functional significance, which is a common approach; however, relying solely on alanine mutations may oversimplify the interpretation of functional roles. They could discuss the potential limitations of this approach and consider a broader range of substitutions (e.g., larger residues or charged residues) to provide more insight into the specific roles of the targeted residues.

Response: Mutating protein residues to Ala is a most widely accepted approach to probing their functional roles. Through an Ala mutation, the entire side-chain of the residue is removed, while preserving the rigidity of the backbone (as opposed to e.g. mutating the residue to a Gly). In our work with other GPCRs, we also used non-Ala mutations, e.g. employed rationally-guided gain-of-function mutagenesis for CCR5 (Dawson *et al.* 2024 doi:10.1101/2023.11.15.567150), disulfide crosslinking for CXCR4 and ACKR3 (Ngo *et al.* 2020 doi:10.1371/journal.pbio.3000656 and Gustavsson *et al.* 2017 /doi:10.1038/ncomms14135), and charge-swap mutagenesis for CXCR4 (Stephens *et al.* 2020 doi:10.1126/scisignal.aay5024). It must be noted, however, that there are 18-19 non-Ala options for mutating every residue of interest; therefore, without a strong reason for targeted introduction of non-Ala mutations, this effort would not be optimal use of resources, especially if mutant evaluation includes quantitative analyses and agonist concentration responses as done in our work. This was the reason we exclusively focused on Ala mutations in our study.

Alternatively, interface mapping could be performed through unbiased techniques, such as saturating mutagenesis or shotgun mutagenesis (e.g. Wescott *et al.* 2016 doi:10.1073/pnas.1601278113). However, these techniques rarely allow for quantitative evaluation using agonist concentration response curves like we did. Moreover, it is known that the interpretation of results of such unbiased studies may be nontrivial: for example, the effect of random mutations on receptor folding and expression may be hard to decouple from their effects

on signal transduction. By contrast, all mutants that we rationally selected and tested in our work expressed comparably to WT CCR9, **Fig. S18**.

Overall, these considerations informed our decision to use model-guided targeted Ala mutagenesis over other techniques, and we hope can convince the Reviewer of the validity (and, really, optimality for the question at hand) of our approach.

Additionally, it could benefit from a more detailed analysis of the specific residues selected for mutation. For example, how do these residues interact with CCL25, and what are their specific roles in the signaling process? The rationale for selecting specific residues for mutation should be better articulated. For example, while the authors group mutations by spatial location and impact, it would be beneficial to explain why certain residues were prioritized for this analysis and how they were hypothesized to influence chemokine binding and signaling

Response: We thank the reviewer for this comment; the rationale was indeed not articulated well. In selecting mutagenesis candidates, we focused on TM domain residues that made strong side-chain contacts with the chemokine in all or most AF2 ensemble models. Receptor N-terminus and extracellular loops were excluded, except Y202^{ECL2}. Additionally, residue positions extensively studied in the context of other receptors and/or known to compromise receptor folding or expression when mutated (e.g. W104^{2.60} and Q303^{7.39}) were deprioritized. Finally, two TM6 residues (Q267^{6.48} and N271^{6.52}) were added for reasons outside of chemokine contacts, explained in the manuscript. In the revision, we added a supplemental figure to explain this rationale, and revised the text accordingly.

8- The text could better clarify the functional implications of the mutations at S207^{5.31} and T208^{5.32}. While the effects of these mutations on binding and signaling are described, a more explicit connection between the observed changes and the underlying molecular mechanisms would enhance understanding.

Response: The text offers the following explanation of the underlying molecular mechanisms:

- “S207^{5.31} is positioned one helical turn above K211^{5.35} and is proximal to **an entirely hydrophobic surface** on CCL25 30's loop (**Fig. 4E-G**). The elimination of S207^{5.31} hydroxyl group via an alanine mutation would **strengthen CCR9 hydrophobic packing against this surface**, which explains the observed concerted increase in chemokine binding and agonism.”
- “We hypothesize that Arr3 recruitment requires this **“most active” state involving the T208^{5.32}-N36 hydrogen bond**, whereas G protein association is permissive to a range of active-like conformations of CCR9, as previously described for other GPCRs. **The loss of T208^{5.32}-N36 hydrogen bonding** in the T208^{5.32}A mutant would therefore **selectively abrogate Arr3 recruitment** with minimal impact on G protein-mediated Ca²⁺ mobilization.”

9- In Figure 1, part D utilizes BW labeling for certain residues, while parts E and F do not employ the same labeling approach. Consistent visualization and labeling across all parts of a figure are crucial for clarity and ease of understanding. Furthermore, I can see this inconsistency in labeling of all figures in the manuscript.

Response: As described above, in the initial submission, we chose to consistently use BW indices in the text but omit them in the figures to avoid unnecessary crowdedness. In the revision, to address the critiques of this Reviewer and also those of Reviewer 2, we added BW indices to all panels on all figures.

10- In parts B, C, and D of Figure 2, residues are presented as balls without any labels. This presentation limits the figure's informational value, as readers cannot relate the individual balls to specific residues. In Figure 8, there is a similar problem. To enhance the clarity and utility of the figure, it is essential to label each ball appropriately. A redesign of this figure to include clear labeling or an alternative presentation that maintains readability would significantly improve its effectiveness.

Response: Figures 2 and 8 have been redesigned to address this comment and a similar comment from Reviewer 2.

11- Parts H, I, and J of Figure 3 are excessively crowded, which significantly hinders the viewer's ability to interpret the interactions effectively. The representation of interactions as dots contributes to the confusion, obscuring the underlying purpose of the figure. To improve clarity and comprehension, I recommend redesigning these sections.

Response: The goal of Panels H-J of Fig. 3 is to show hydrogen-bonding networks of the three basic residues studied in the corresponding sections. Rather than indicating generic "interactions", the cyan dotted lines indicate hydrogen bonds which are critical for conveying the message of the figure. To make this and other figures clearer, we added BW indices to residue labels on all panels.

Responses to critiques from Reviewer #5

The manuscript describes a hybrid computational experimental approach to elucidating mechanisms of CCR9-CCL25 interaction and CCR9 activation and provides insight in structure based drug discovery. Overall approach is sound and it is nice that the results of computational modeling are followed by systematic mutational studies to support the model. The manuscript is well written and easy to follow and provides a lot of useful information on the important system. Below are a few comments which might be useful to include in the final version of the manuscript. I believe extended analysis of the models /reporting interface confidence and providing analysis of models in the supplement together with reporting the final model together with overlapped experimental results would greatly improve the manuscript.

Response: We are thankful to the reviewer for their positive assessment of our work and agree that additional model analyses are helpful. We included the requested data in the revised version of the manuscript. Point-by-point responses to the comments are provided below.

1) It would be good to store/report somewhere final suggested model in pdb and potentially pse (pymol session) format with mutated residues highlighted and color coded for the agreement with experimental data . Maybe I missed it somewhere but I can't see it anywhere.

Response: For the initial submission, we provided the folder with PDB files as supplementary data, with the quantitative fields used for model coloring in the B-factor field of the models. With this revision, we also provide interactive molecular graphics sessions for all figures in the .icb format (ICM binary, can be opened and manipulated in the free ICM Browser).

2) Alternative models are also potentially interesting to report together with data to see the level of uncertainty.

Response: These models were provided as part of the PDB supplementary data for the initial submission and are also included in the revision. Also, the entire 25-model ensemble is shown in **Figs. 4H-J**.

3) I see the authors used their own confidence model for the choice of the model - what was the original AF2 Multimer confidence at the interface of the best and selected model.

Response: The AF2-generated pLDDT scores for residues 20-340 of CCR9 and residues 1-80 of CCL25 are now reported in **Fig. S1**, and for all CCR9 CRS2 residues in **Fig. S6B**. For all models, the scores in CRS2 exceeded 80, and for most residues exceeded 90 (i.e. very high confidence), justifying the use of alternative metrics (like RTCNN) for model discrimination and in-depth studies of model features. Uncertainties were only identified in the receptor N-terminus and (to a lesser extent) in ECL2, i.e. outside of the regions we focused on in the study.

Predictions for the chemokine side were similarly highly confident. An exception was the distal N-terminus, where lower pLDDT scores were observed: 45-57.5, 50-56, and 58-75 for the chemokine N-terminal residues 1, 2, and 3, respectively (**Fig. S1**). This was associated with larger conformational variation of these residues in the AF2-generated ensemble (**Fig. S2-S4**). Despite the variation, the key intramolecular contacts (e.g. that of chemokine pGlu1 and CCR9 Y126^{3,32}) were preserved in all models. The lower prediction confidence and the higher conformational variation of the chemokine N-terminus is consistent with its minor role in CCR9 binding and activation which is experimental confirmed by our study (**Fig. 6**). Note that conformational variability is an inherent part of GPCR-ligand complexes and is frequently observed not only in models but also among experimental structures (e.g. Note that conformational variability is an inherent part of GPCR-ligand complexes and is frequently observed not only in models but also among experimental structures (e.g. Michino et al., 2009 doi.org/10.1038/nrd2877; Lane et al., 2023 doi.org/10.1038/s41592-022-01760-4). In our manuscript, for the analyses focusing on specific parts of the receptor-chemokine interface, models with highest confidence in those parts were selected from the ensemble (e.g. model rank 0 for **Fig. 5**).

4) The authors run 5 seeds/5 models (ensemble of 25 structures) for AF2 - while it could be enough for easy cases - I would run 100 seeds 5 models (as a matter of fact this is what community finds the best as well as DeepMind itself) - would be good to confirm that most confident structure doesn't change.

Response: Done. In the revision, we included **Figs. S2-S4** that demonstrate the levels of conformational variation within our original ensemble of 5x5 AF2 models as well as between these models and the top 20 most confident models from the newly built 100x5 AF2 ensemble as proposed by the reviewer. The larger AF2 ensemble provides no improvements in model confidence. Conformationally, the most confident models from the larger AF2 ensemble are highly similar (<2Å RMSD) to the models we present in the manuscript in terms of the receptor's binding pocket and the chemokine's 30s loop. Consistent with lower confidence for the prediction of the chemokine N-terminus, there is variation in its conformations both within the ensembles and between the ensembles. Despite this variation, all key intermolecular contacts studied in the paper are preserved.

5) Since AF3 server is available I would also suggest to run multiple seeds of AF3 - may be 20 (as currently available from single account)

Response: Done. The above-mentioned **Figs. S2-S4** also include the comparisons of the original ensemble of 5x5 AF2 models to the 20x5 newly constructed AF3 models as suggested by the reviewer. The AF3 ensemble features the same level of confidence as the AF2 ensembles in the

receptor's binding pocket and the chemokine's 30s loop; however, in the chemokine N-terminus, we observe an improved confidence and less conformational variation in the AF3 models compared to AF2 models. The contacts studied in the paper are preserved. Interestingly, AF3 models tend to be more active-like, according to our metric introduced in **Fig. 4**, and more frequently feature the H-bond between CCR9 T208^{5.32} and CCL25 N36/o, i.e. these new models confirm our earlier predictions regarding the structural basis for full vs partial receptor activation. Note that the AF3 Terms of Use prevent us from repeating the analyses presented in the paper on this new set of models. However, we now include the 20x5 AF3 models, in the .pdb format, as **Supp. Data 2**.

POINT-BY-POINT RESPONSES TO REVIEWERS' COMMENTS

Reviewer #2 (Remarks to the Author):

We thank the authors for their thorough and thoughtful responses to the comments from all reviewers.

The revised manuscript, 'Noncanonical roles of chemokine regions in CCR9 activation revealed by structural modeling and mutational mapping,' stands out as one of the few studies to rigorously validate a computational model with experimental approaches. This validation and the novel insights into the CCL25 binding mode make it a strong contribution to the chemokine field.

Following careful review, we find that the authors have effectively addressed the limitations of the original manuscript through detailed and constructive responses. We recommend this manuscript for publication.

Our only remaining recommendation is that the authors include the raw calcium flux and BRET assay data to enhance reproducibility and transparency. We also suggest that the authors include the reviewer responses as supplementary material, as they provide important insights into the methodology and its inherent limitations.

Response: We thank the reviewer for this feedback. As requested, we have ensured that all raw data underlying the calcium flux and BRET assay results are provided as source data files. Regarding the suggestion to include reviewer responses as supplementary material, we understand that *Nature Communications* offers a transparent peer review option that would make these exchanges publicly available alongside the published article. We are supportive of this practice and would be happy for our peer review file to be published, should the journal proceed in this direction.

Reviewer #3 (Remarks to the Author):

Reviewer #4 (Remarks to the Author):

Thank you for the revisions. The manuscript is now more clearly presented in terms of methodology, and the interpretation of the results has improved with a more balanced discussion, particularly in relation to the limitations of AlphaFold2. The inclusion of a brief acknowledgment regarding the restricted ability of AlphaFold2 to model dynamic aspects of GPCR activation helps contextualize the findings, though this remains an inherent limitation of the computational approach.

The mutagenesis and structural modeling are well-integrated, and the logic behind residue selection is now better explained. However, the absence of direct experimental validation of predicted contacts still leaves room for ambiguity. While you address this by citing prior experience and noting the technical challenges, it would be useful to be more explicit in the manuscript about how future work might aim to close this gap. For example, stating whether charge-reversal or crosslinking strategies are being considered or how feasible they might be in this system could help orient readers and frame the current findings as part of an ongoing effort rather than a complete resolution.

In its current form, the manuscript is more solid, but this clarification would improve the framing of the conclusions and more transparently communicate the limitations and opportunities ahead.

Response: We thank the reviewer for this feedback. In response to the suggestion to clarify how future work might address the absence of direct experimental validation of predicted contacts, we have amended the fifth paragraph of the Discussion to explicitly mention potential charge-swap and disulfide crosslinking approaches, along with the broader need for experimental structure determination. We agree that these represent important avenues for future study and hope the revised text provides a clearer framing of the current findings as part of an ongoing effort.

Reviewer #5 (Remarks to the Author):

The authors did a great job on the revision. All requested additional runs with AF2 and AF3 are done , and deposited. I think the manuscript is good now from my viewpoint.

Response: We thank the reviewer for this positive assessment.